# Contribution of classical end-joining to PTEN inactivation in p53-mediated glioblastoma formation and drug-resistant survival

Youn-Jung Kang[1,2], Barbara Balter[1], Eva Csizmadia[3,4], Brian Haas[2], Himanshu Sharma[1], Roderick Bronson[5] & Catherine T. Yan[1,2]

DNA repair gene defects are found in virtually all human glioblastomas, but the genetic evidence for a direct role remains lacking. Here we demonstrate that combined inactivation of the *XRCC4* non-homologous end-joining (NHEJ) DNA repair gene and p53 efficiently induces brain tumours with hallmark characteristics of human proneural/classical glioblastoma. The murine tumours exhibit PTEN loss of function instigated by reduced *PTEN* mRNA, and increased phosphorylated inactivation and stability as a consequence of aberrantly elevated CK2 provoked by p53 ablation and irrevocably deregulated by NHEJ inactivation. This results in DNA damage-resistant cytoplasmic PTEN and CK2 expression, and the attenuation of DNA repair genes. CK2 inhibition restores PTEN nuclear distribution and DNA repair activities and impairs tumour but not normal cell survival. These observations demonstrate that NHEJ contributes to p53-mediated glioblastoma suppression, and reveal a crucial role for PTEN in the early DNA damage signalling cascade, the inhibition of which promotes tumorigenicity and drug-resistant survival.

[1] Department of Pathology, Beth Israel Deaconess Medical Center, Boston, Massachusetts 02215, USA. [2] The Broad Institute of MIT and Harvard, Cambridge, Massachusetts 02143, USA. [3] Department of Surgery, Beth Israel Deaconess Medical Center, Boston, Massachusetts 02215, USA. [4] Transplant Institute, Beth Israel Deaconess Medical Center, Boston, Massachusetts 02215, USA. [5] Dana-Farber/Harvard Cancer Center Rodent Histopathology Core, Harvard Medical School, Boston, Massachusetts 02215, USA. Correspondence and requests for materials should be addressed to C.T.Y. (email: cyan@bidmc.harvard.edu).

Glioblastoma multiforme (GBM) is classified as a grade IV diffuse high-grade astrocytic glioma (HGA). It is the most common adult brain tumour and has uniformly poor outcome largely because of resistance to conventional surgical and genotoxic therapies[1,2]. Mutations in the components of the p53, PTEN/PI3K/AKT (phosphatase and tensin homologue/phosphatidylinositol-3 kinase/AKT) or RB (retinoblastoma) pathways are found in the majority of human primary GBMs, with p53 as the most frequently mutated gene in low-grade and secondary GBMs[3,4]. In mouse models, no evidence of brain tumours was detected in germline p53, RB or PTEN single knockouts[5]. Despite the frequent co-occurrence of p53 and PTEN gene defects in human GBMs, GBMs represent only a small fraction of the HGAs induced by GFAP-CreER in p53/PTEN and p53/PTEN/RB knockout mice[5], and hGFAP-Cre in p53[Ex5-6DEL] mice[6]. These findings and additional evidence of PTEN (80%) and/or NF1 (14%) protein loss in p53[Ex5-6DEL] HGAs demonstrated an important role for p53 gene defects in the early stages of glioma development[6]. Cumulatively, these observations support the view that additional genetic alterations are required to drive p53-mediated gliomagenesis.

We and others demonstrated that germline or tissue-specific knockout of p53 contributes to cancer development in various DNA repair-deficient backgrounds, in particular, in mice deficient in XRCC4 and DNA ligase IV (LIG4), key components of the non-homologous end-joining (NHEJ) DNA repair pathway[7,8]. Several studies tied functional polymorphisms in XRCC4, LIG4 and other DNA repair pathway genes to increased risk of GBM[9–13]. NHEJ gene defects are reportedly found in a broad spectrum of human cancers including GBMs with p53 aberrancies[14], but evidence for a direct role for NHEJ in GBM pathogenesis remains lacking.

NHEJ is the major pathway in mammalian cells that repairs DNA double-strand breaks (DSBs) resulting from normal metabolism or induced by clastogens such as ionizing radiation[15]. Knockout of any of components of the canonical NHEJ machinery including XRCC4, LIG4, DNA-PKCs, KU70, KU80, ARTEMIS and XLF in mice results in severe cellular radiosensitivity, genomic instability, impaired proliferation and survival[15]. However, NHEJ knockout mice are not particularly cancer prone because of intact p53-mediated apoptotic elimination of irreparably damaged cells[7,16].

One of the major critical challenges in GBM pathogenesis is in defining the critical steps driving the development of this complex disease. In this study, we develop a murine GBM model in which knockout of both p53 and XRCC4 is specifically targeted in neural stem/progenitors in the developing brain, and use this model to investigate the impact of NHEJ deficiency on p53-mediated gliomagenesis.

## Results

### XRCC4/p53 deficiency transforms only glioblastoma precursors.
To investigate the relevance of NHEJ gene defects in human GBMs, we assessed the data from human GBM samples in the COSMIC[17] and cBioPortal database[14] for molecular alterations in NHEJ genes. This revealed that nearly 14% (97/695) were reduced in NHEJ gene expression (XRCC4 (8/695), LIG4 (19/695), Ku80/XRCC5 (6/695), Ku70/XRCC6 (44/695), Artemis/DCLRE1C (18/695) and Cernunnos/NHEJ1 (2/695)) (Supplementary Table 1 and Supplementary Fig. 1a). XRCC4 and p53 (TP53) gene defects frequently co-occurred with a P value of 0.0208 (Supplementary Fig. 1b). Analysis of 32 accessible data from the 97 samples revealed genomic alterations in other GBM-relevant genes besides TP53 (14/32), including PTEN (15/32), RB1 (6/32), CDKN2A/B

(21/32), EGFR (20/32), PIK3CA (2/32) and/or AKT3 (2/32) (Supplementary Table 1).

Previously, we found that rare HGAs characteristic of GBMs were induced by simultaneous ablation of XRCC4 with p53 using CD21-Cre[18] that limits Cre transmission to peripheral B cells[19] and to GFAP[+] radial astrocyte-derived adult subgranular zone neural stem (NS) cells in the dentate gyrus[20]. Here, to directly assess the contribution of NHEJ in p53-mediated gliomagenesis, we interbred XRCC4 and p53 conditional knockouts[21] to transgenic mice in which Cre recombinase transmission is driven by human glial fibrillary acidic protein (hGFAP-Cre) promoter expression in embryonic radial glial cells and in mature neurons and astrocytes[22] (Fig. 1a and Supplementary Fig. 1c,d). hGFAPCreX[fl/fl]p[fl/fl] (GXP) and cohort XRCC4-deleted (hGFAPCreX[fl/fl]; GX), p53-deleted (hGFAPCreP[fl/fl]; GP) and hGFAPCreX[fl/fl]p[fl/+] (GXP[het]) mice were monitored up to 17 months of age. Significantly, although none of the cohort GX, GP and GXP[het] mice developed tumours (Fig. 1b), GXP mice (n = 37) developed brain tumours with 100% penetrance. Of these, 48.6% (18/37), detected between 100 and 160 days of age, were classified as medulloblastoma (Fig. 1b; hGFAPCreX[fl/fl]p[fl-MB], Supplementary Fig. 1ei,ii). The remaining 51.4% (19/37), detected between 8 and 16 months of age, were all histologically classified as GBMs (Fig. 1b; hGFAPCreX[fl/fl]p[fl-GBM]; hereafter referred to as GXP GBM) based on diffuse forebrain (Fig. 1c) and spinal cord infiltrations (Supplementary Fig. 1eiii,iv); marked cellular pleomorphism, pseudopalisading necrosis (Fig. 1ci,ii) and nuclear atypia (multinucleated giant cells, black arrows in Fig. 1ciii); and high mitotic indices (evidenced by elevated proliferating-cell nuclear antigen (PCNA); Fig. 1d). Typical of human astrocytic tumours, they expressed high levels of GFAP (Fig. 1d), and rarely the neuronal marker, NeuN (Supplementary Fig. 1f). Significantly, this is the first mouse model in which HGAs are only represented by GBMs. In contrast, GBMs represented only 25% of the HGAs induced in p53/PTEN and p53/PTEN/RB knockout mice,[5] and 40% of the HGAs in p53[Ex5-6DEL] mice in which medulloblastomas were also induced[6]. In adult GXP (also CD21-CreX[f/f]p53[f/f] ref. 18) mice, it may be that simultaneous ablation of XRCC4 and p53 provides selective growth advantage to very limited pool of NS during adult neurogenesis that, based on the intersection of adult NS that express both hGFAP-Cre and CD21-Cre[20], may originate from the adult subgranular zone.

### GXP GBMs exhibit proneural-classical GBM signatures.
To molecularly classify the GBMs induced in GXP mice, we performed array comparative genomic hybridization (aCGH) on 17 tumours. This revealed most carried complex copy number (CN) alterations commonly found in human GBMs[3,14], including amplifications (CN > 3) of CCND2, CCND1, AKT3, MET, EGFR, CDK6, PIK3CA, PDGFRA, CMYC and/or MYCN, and/or deletions (CN < 1.5) of CDKN2A/B/C, PTEN and/or RB1 (Supplementary Tables 1 and 2 and Supplementary Fig. 1g). Spectral karyotyping further revealed that murine GBMs, like in humans, are highly aneuploid and harbour clonal, nonrecurrent translocations (Supplementary Fig. 1h). Comparison of RNA sequencing (RNA-seq) of 6 GXP GBMs to genetically matched postnatal day 1–3-derived NS progenitors, and a 840 gene list used to classify human GBM subtypes[3,4] revealed all 6 GXP GBMs to recapitulate the molecular signatures of proneural GBMs, and to a lesser extent classical GBMs (Fig. 1e). These GBM subtypes are hallmarks, respectively, of secondary/low-grade gliomas associated with p53 aberrancies[4], and primary HGAs associated with PTEN aberrancies[3,4]. Indeed, our interrogation of human GBMs in cBioPortal[14] discovered that PTEN mRNA is concordantly reduced with molecular alterations in both p53 and

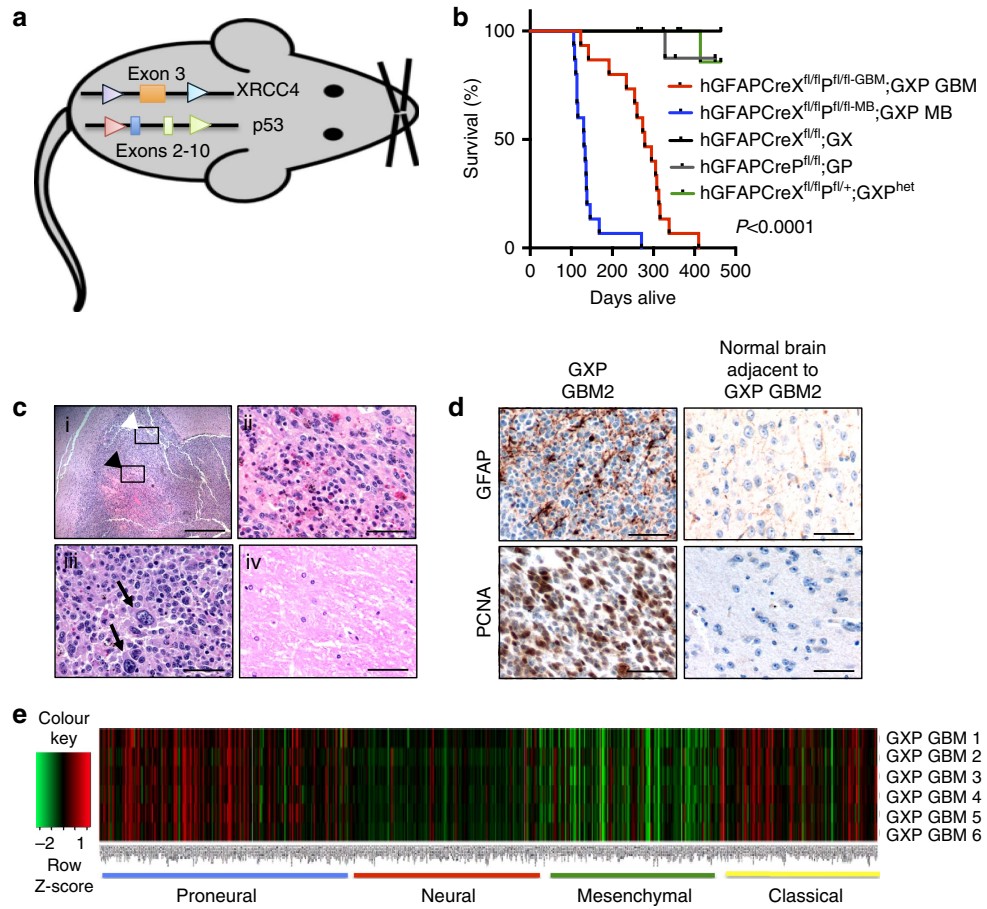

**Figure 1 | Configuration and characterization of GBMs developed from hGFAP-CreXRCC4<sup>fl/fl</sup>p53<sup>fl/fl</sup> mice.** (**a**) Schematic diagram of loxP-flanked *XRCC4* and p53 conditional knockout alleles, bred into hGFAP-Cre transgenic mice. (**b**) Kaplan–Meier survival analyses on hGFAPCreX<sup>fl/fl</sup>P<sup>fl/fl-GBM</sup> (GXP GBM; *n* = 19), hGFAPCreX<sup>fl/fl</sup>P<sup>fl/fl-MB</sup> (GXP MB; *n* = 18), hGFAPCreX<sup>fl/fl</sup> (GX; *n* = 29), hGFAPCreP<sup>fl/fl</sup> (GP; *n* = 11) and hGFAPCreX<sup>fl/fl</sup>P<sup>fl/+</sup> (GXP<sup>het</sup>; *n* = 7) mutant mice. Data are analysed using log-rank (Mantel–Cox) test. (**c**) Haematoxylin and eosin (H&E) staining of serial sections of a representative primary GXP GBM exhibiting marked cellular pleomorphism, pseudopalisading necrosis (ii; magnified of black arrowhead box indicated in i) and multinucleated giant cell phenotypes (iii; black arrows; magnified of white arrowhead box indicated in i), compared with its adjacent normal brain tissue (iv). Scale bars, 200 μm (i) and 50 μm (ii–iv). (**d**) Immunohistochemical staining of GXP GBM2 and its adjacent normal brain tissue with antibody against GFAP or PCNA. Scale bars, 50 μm. (**e**) Unsupervised hierarchical clustering analysis of RNA-seq. Each row represents a distinct sample (6 independent GXP GBMs compared with GXP NS) and each column represents an individual gene (based on a list of 840 genes that classifies the four TCGA human GBM subtypes). Normalized (log2) and standardized (each sample to mean signal = 0 and s.d. = 1) level of gene expression is denoted by colour (green; low, dark; intermediate, red; high), as indicated in the gradient panel. See also Supplementary Fig. 1.

*XRCC4* (*P* = 0.0471). Quantitative reverse transcription polymerase chain reaction (qRT–PCR) analysis revealed *PTEN* mRNA is significantly reduced and PI3K/AKT pathway genes are correspondingly activated in all 17 GXP GBMs (Fig. 2a and Supplementary Fig. 2a). To validate the induction of PI3K/AKT pathway proteins, we performed immunoblot analysis, examining the expression levels of PIK3CA, PIK3CB and p-AKT in seven representative independent GXP GBMs. This revealed that levels of PIK3CB and p-AKT (S473) were elevated in all seven GXP GBMs and GBM cell lines we tested compared with wild-type (WT) NS or WT brain, whereas the levels of PIK3CA and p-AKT (T308) appeared to be more heterogeneously expressed (Fig. 2b). Correspondingly, treatment of tumour cells in cultures with PI3K (LY294002)[23] or AKT (MK2206)[24] inhibitors substantially impaired their survival (Supplementary Fig. 2b,c). Overall, these observations definitively demonstrate that deregulation of PI3K/AKT signalling contributes to their malignant progression.

**Reduced *PTEN* mRNA is countered by CK2 in GXP GBMs.** To determine whether PTEN protein levels are concordant with the

reduction in *PTEN* mRNA in GXP GBMs, similar to the observation in 80% of p53<sup>Ex5-6DEL</sup> gliomas[6], we performed anti-PTEN immunofluorescence (IF) staining on paraffin-embedded primary GBM-bearing brain sections. This revealed two unexpected alterations. In the tumour cells, PTEN protein levels were comparable to those observed in adjacent normal tissue or WT adult brain, and PTEN is predominantly cytoplasmic instead of being nuclear (colocalized with PCNA as in normal tissue) (Fig. 2c). These alterations are consistent with increased PTEN stability that can be provoked by phosphorylation of its C terminus at S380/T382/T383 (and flanking residues) by the protein kinase casein kinase 2 (CK2)[25]. Correspondingly, the levels of S380/T382/T383-phospho-PTEN (p-PTEN) and CK2 via its β-subunit of *CSNK2B* (CK2β) were elevated in GXP GBMs, and in the murine K-ras/p53 mutant GL261[26], human p53/PTEN mutant T98G[27] and human PTEN-null U87MG[27] GBM cell lines as compared with control NS and WT adult brain (Fig. 2d and Supplementary Fig. 2d). Notably, CK2β protein is modestly induced in GXP NS and nearly undetectable in X<sup>fl/fl</sup>P<sup>fl/fl</sup> (XP-undeleted; WT) NS and normal adult brain (Fig. 2d), indicating CK2 activity

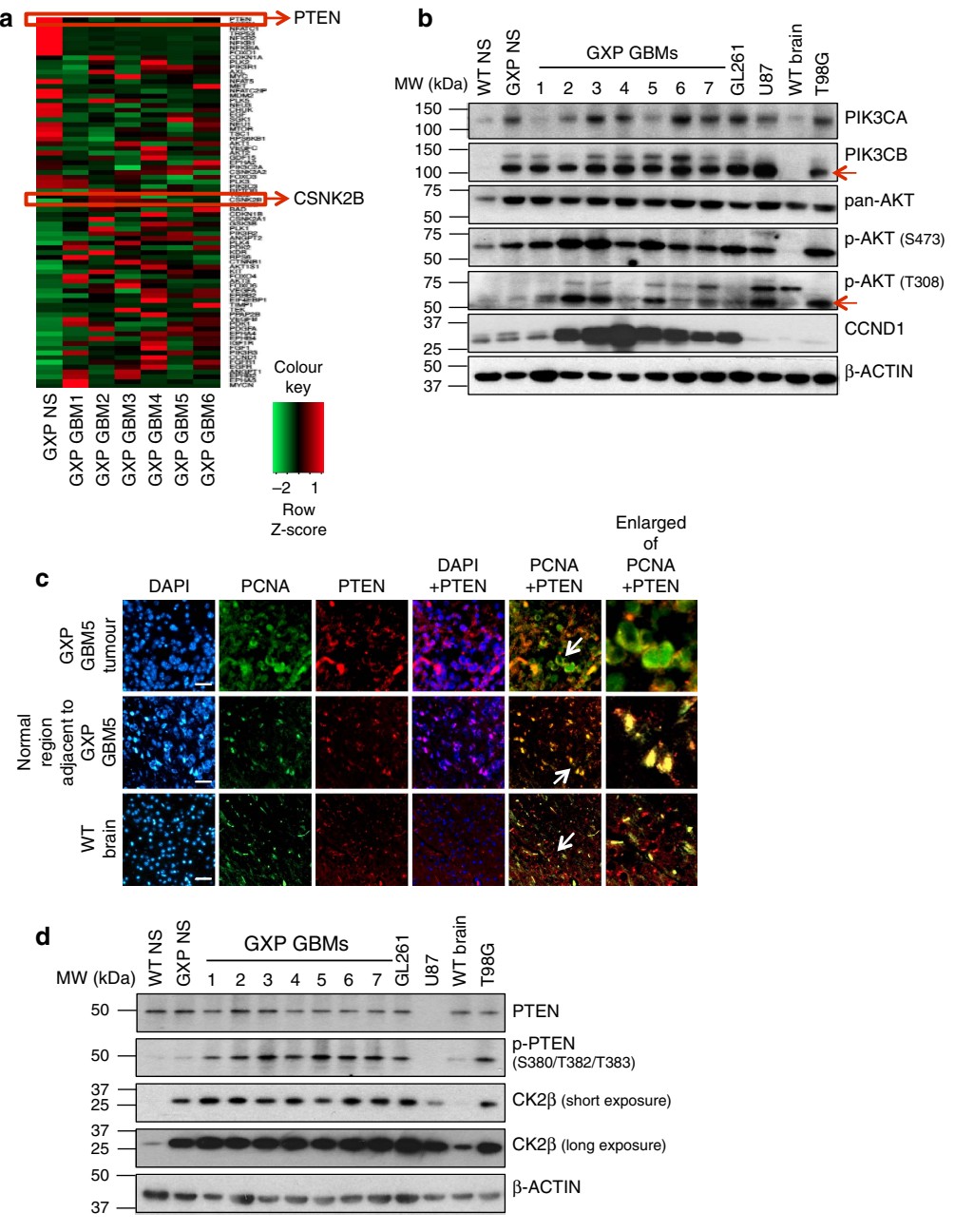

**Figure 2 | PTEN loss of function in GXP GBMs.** (**a**) Heatmap of the PI3K/AKT signalling pathway genes generated from RNA-seq analysis highlighting downregulated expression level of PTEN and upregulated expression level of CSNK2B (CK2β) in 6 independent GXP GBMs compared with GXP NS. (**b**) Immunoblot analysis of PIK3CA, PIK3CB (indicated by red arrow), pan-AKT, p-AKT (S473), p-AKT (T308) (indicated by red arrow) and CCND1 in 7 independent GXP GBMs compared with WT or GXP NS, WT adult forebrain, GL261 (p53-mutated), U87MG (PTEN-null) or T98G (p53/PTEN mutated). MW; molecular weight. (**c**) Co-IF staining of PTEN and PCNA on primary GXP GBM5 paraffin-embedded sections, tumour cells (top panel) compared with adjacent normal tissue (middle panel) and WT adult brain tissue (bottom panel) are shown. PCNA and DAPI staining of nuclei are shown. White arrows in PCNA + PTEN merged images denote the regions enlarged in the far-right panels. Scale bars, 50 μm. (**d**) Immunoblot analysis of PTEN, p-PTEN (S380/T382/T383) and CK2β in seven independent GXP GBMs compared with WT or GXP NS, WT adult forebrain, GL261, U87MG or T98G. See also Supplementary Fig. 2.

may be instigated by DNA damage induced in the XRCC4/p53-deficient setting and pathologically altered in GXP GBMs and the tested GBM cell lines. To assess PTEN protein stability, GBMs and control NS were pulse-chased in culture with the protein synthesis inhibitor cyclohexamide[28], and its turnover rate was followed in the absence or presence of CK2-specific inhibitor tetrabromobenzotriazole (TBB)[29]. This revealed a slower rate of PTEN turnover in both GXP and T98G GBMs that could be restored to near physiological levels by TBB (Supplementary

Fig. 2e,f), definitively demonstrating that their reduced *PTEN* mRNA levels are countered by CK2. Our observations here, together with findings from previous p53 HGA models[5,6], are consistent with PTEN loss of function as a central factor driving p53-mediated gliomagenesis.

**DNA damage-resistant cytosolic CK2 and PTEN retention.** PTEN accumulates in the nucleus upon DNA damage[30,31]. The steady-state primarily cytoplasmic PTEN expression in GXP

GBMs *in vivo* (Fig. 2c), like in human sporadic melanoma and thyroid carcinoma[32], led us to next ask whether DNA damage-induced PTEN nuclear expression along with CK2β might pathologically be altered. GBM cells and relevant controls were treated with the radiomimetic DNA damage-inducing drug doxorubicin[33]. In the absence of doxorubicin in culture, both PTEN and CK2β were primarily cytoplasmic in both GBMs and control NS. Following 5 h of doxorubicin exposure, both became predominantly nuclear in control NS, and remained cytoplasmic in GXP GBMs and T98G (Fig. 3a and Supplementary Fig. 3a). Immunoblot analysis showed that cytoplasmic PTEN in GXP GBMs is predominantly phosphorylated (Fig. 3b), and neither CK2β nor PTEN protein levels are affected by doxorubicin (Fig. 3c). These observations demonstrate that PTEN and CK2 status are somehow coordinately pathologically altered in p53-mutant/deficient GBMs.

**PTEN-CK2 status and delayed DNA damage signalling.** Increased DNA damage response (DDR) signalling and DNA repair (DDR/repair), based on elevated expression of certain associated enzymes in human GBM cells, is suggested to contribute to their ability to survive high levels of DNA damage[34,35]. To address this, DDR/repair dynamics in GXP GBMs compared with T98G and control NS were assessed. Cultured cells were treated with doxorubicin to induce DNA damage. DDR/repair kinetics was examined at 0, 0.5, 1 and 5 h of doxorubicin exposure and at 5 or 24 h after treatment, by following the formation of DNA damage-induced γH2AX foci that assembles DNA repair proteins at DSBs[36], and the recruitment of RAD51 or 53BP1 foci, surrogate markers widely used respectively to assess the activities of homologous recombination (HR) and NHEJ[36]. Notably, RAD51 foci were rarely detected in NS controls and GXP GBMs (Supplementary Fig. 3b), indicating HR is not a dominant mechanism despite the deficiency in NHEJ. Unexpectedly, 53BP1 foci were readily induced in the XRCC4-deficient GX NS demonstrating 53BP1 also marks the recruitment of non-NHEJ DNA repair proteins to DSBs (Fig. 3d,e and Supplementary Fig. 3c). Significantly, DDR kinetics is severely delayed in GXP GBMs and T98G in which γH2AX/53BP1 foci were detected only after 5 h of doxorubicin exposure, as compared with 0.5 h for WT and GX NS, and 1 h for GP and GXP NS (Fig. 3d,e and Supplementary Fig. 3c). Because of the severe delay in DDR kinetics in the GBMs, the kinetics of DSB repair was assessed at 5 or 24 h of recovery after 5 h of doxorubicin exposure, instead of at earlier time points. This revealed rapid γH2AX/53BP1 foci co-dissociation in WT, GX, GP or GXP NS that failed to dissolve in both GXP GBMs and T98G even at 24 h of recovery (Fig. 3d,e and Supplementary Fig. 3c). These data demonstrate broad attenuation of DDR/repair in the GBMs that appears to correlate with coordinated altered distribution of both PTEN and CK2β. This may be because CK2 regulates the activities of DSB repair proteins including XRCC4 and XRCC1[37,38], and like PTEN, redistributes to the nucleus following genotoxic stress[39]. Correspondingly, stress-induced nuclear p-PTEN activity augments p53-mediated $G_1$ cell cycle arrest and reduces p53-mediated ROS production[40].

Therefore, to test whether pathological CK2 activity and PTEN subcellular distribution are coordinately regulated, we treated GXP GBMs, T98G and control NS with TBB. As shown above, by competitively inhibiting CK2 kinase activity, TBB reduced CK2-mediated PTEN phosphorylation and restored PTEN protein turnover rate (Supplementary Fig. 2e,f) to reflect its physiologically reduced mRNA levels in GXP GBMs (Supplementary Fig. 2a). Indeed, immunoblot analysis revealed that TBB reduced CK2, total PTEN and p-PTEN protein levels, and consequently decreased p-AKT and p-PRAS40 in GXP GBMs (Fig. 4a). This latter effect is most likely a direct consequence of negative regulation of PI3K/AKT signalling by TBB-restored catalytically active but reduced PTEN levels, although alternative mechanisms including a direct impact of TBB on AKT activation have been proposed[41,42]. By IF and nuclear/cytoplasmic fractionation, TBB increased PTEN nuclear expression, although a substantial fraction still resided in the cytoplasm, and further restored DNA damage-induced nuclear expression of the remaining PTEN protein in GXP GBMs and T98G, whereas nuclear PTEN level was only partially restored in GXP NS, and minimally in WT NS (Fig. 4b and Supplementary Fig. 4a,b). Importantly, co-IF quantification of γH2AX/53BP1 foci showed TBB resensitized GXP GBM and T98G cells to doxorubicin-induced DNA damage and restored DNA repair activities to the levels observed in GXP NS (Fig. 4c,d and Supplementary Fig. 4c), demonstrating coordinated regulation of physiologic PTEN in DDR/repair activity. RNA-seq transcriptome analysis of the GXP GBMs identified 6 affected DNA repair factors (*PARP1*, *FANCD2*, *RAD50*, *MRE11A*, *LIG3* and *RBBP8*) with implicated roles in HR and other DNA repair pathways, including the NHEJ-independent alternative end-joining (A-EJ) DSB repair pathway[43,44]. Analysis of their expression by qRT–PCR in cultured cells in the absence or presence of TBB revealed that TBB treatment significantly increased the mRNA levels of all 6 HR/A-EJ-related genes in GXP GBMs and T98G, and not in WT or GXP NS (Supplementary Fig. 4d). To corroborate these findings, we asked whether FANCD2, which binds to DSB intermediates like NHEJ, is induced in TBB-treated cells upon doxorubicin-induced DNA damage[43,44]. IF analysis revealed that exposure of TBB-treated GXP GBMs to doxorubicin resulted in significantly higher numbers of cells positive for FANCD2 foci (with ≥ 5 foci per cell) compared with dimethylsulfoxide-treated GXP GBMs (Fig. 4e,f). Moreover, longer exposure to TBB impaired the growth and survival of GXP GBM cells and had little impact on NS controls including GXP NS (Supplementary Fig. 4e,f), indicating only the pathologic status of CK2 is targeted. Cumulatively, these data demonstrate that loss of DDR/repair activities in GBMs is correlated to reduction in unphosphorylated nuclear PTEN and the pathologic status of CK2. Significantly, they suggest that a combinational anticancer therapy utilizing a CK2-specific inhibitor before induction of genotoxic stress could represent an effective treatment for killing DNA damage-resistant GBMs and sparing noncancerous counterparts.

**Overexpressing CK2β transforms GXP NS by impairing PTEN.** We next asked whether overexpressing CK2β enhances the cellular transformation of control NS, particularly GXP NS, the progenitors from which the murine GBMs are derived and in which CK2β is moderately expressed. GXP, GX, GP and WT NS were transduced with lentivirus expressing cytomegalovirus haemagglutinin (CMV-HA)-tagged CK2β cDNA that could be distinguished from endogenous CK2β by anti-HA detection (Fig. 5a and Supplementary Fig. 5a). This resulted in elevated p-PTEN and p-AKT (Fig. 5a), rapid proliferative growth of GXP NS (Fig. 5b–e) and moderately increased the proliferation of GP NS but not of GX NS or WT NS (Supplementary Fig. 5b and Fig. 5d). However, whereas PTEN is predominantly redistributed to the nucleus in GX, GP and GXP NS in response to doxorubicin-induced DNA damage, a substantial fraction of PTEN remained cytoplasmic in doxorubicin-treated CK2β-transduced GP and GXP NS (Fig. 5f). In contrast, doxorubicin treatment of CK2β-transduced GX NS resulted in a normal pattern of DNA damage-induced PTEN nuclear expression (Fig. 5f). These data suggest that the pathologic alterations in CK2 and PTEN may be enforced by

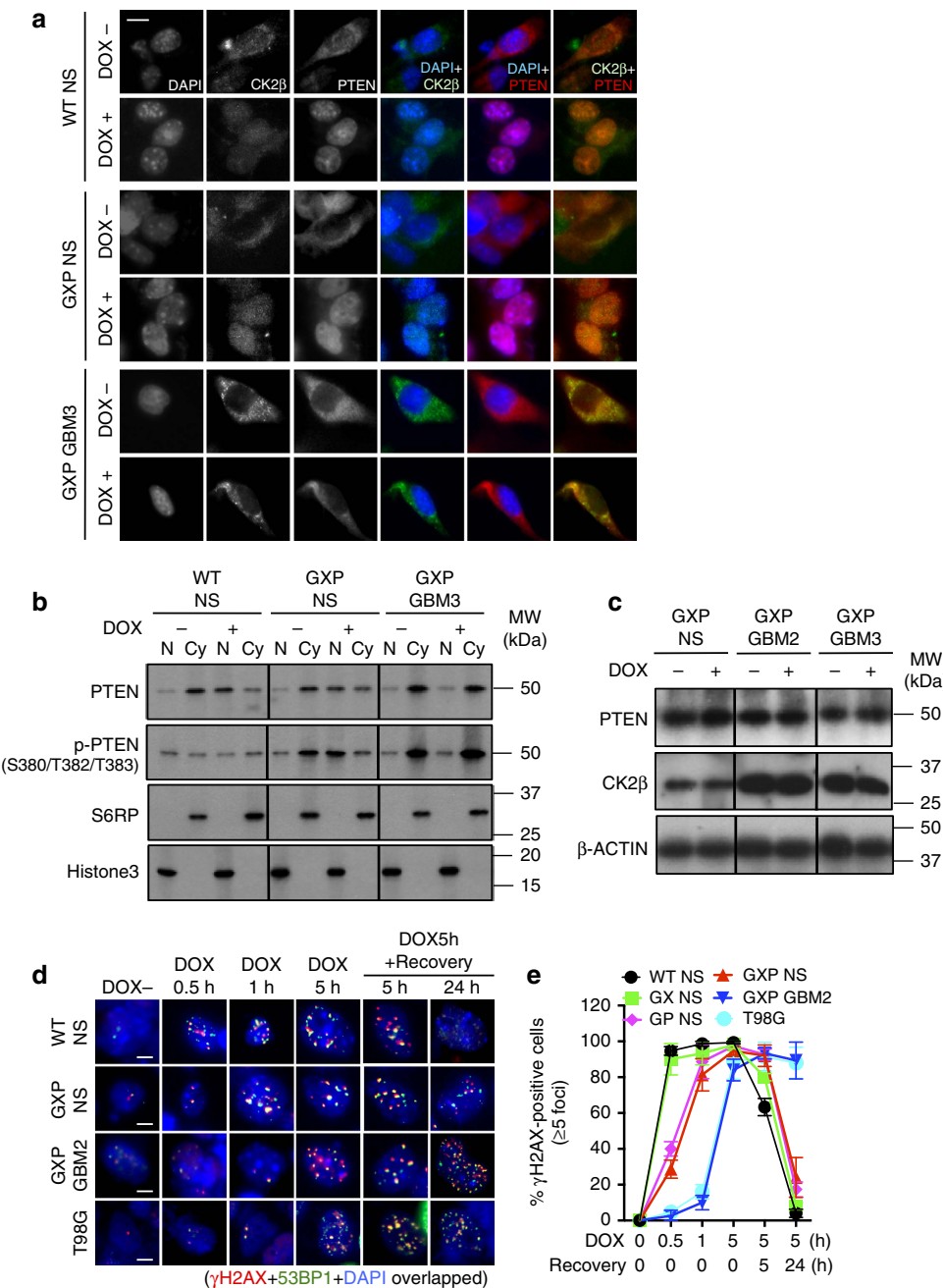

**Figure 3 | Correlation of CK2 and PTEN subcellular distribution with delay in DNA damage response in GBMs.** (**a**) Co-IF of CK2β and PTEN with DAPI nuclear staining in the absence of doxorubicin (DOX −; dimethylsulfoxide (DMSO)-treated) or after 5 h of exposure to DOX (DOX +; 0.5 μM DOX-treated) in GXP GBM3 compared with WT or GXP NS. Representative images are shown. See Supplementary Fig. 3a for representative IFs of GX or GP NS, GXP GBM2 and T98G. Scale bar, 20 μm. (**b**) Immunoblot analysis of nuclear (N) and cytosolic (Cy) distributions of total PTEN and S380/T382/T383-phospho-PTEN in WT or GXP NS, or GXP GBM3 in DOX − (DMSO-treated) and DOX + (0.5 μM/5 h) conditions. Loading controls: S6RP (cytosolic) and Histone3 (nuclear). (**c**) Immunoblot analysis of PTEN and CK2β in two representative GXP GBMs compared with GXP NS in DOX − (DMSO-treated) and DOX + (0.5 μM/5 h). Loading control: β-actin. (**d**) Co-IF staining of DOX-induced foci of γH2AX and 53BP1 in WT or GXP NS, GXP GBM2 and T98G. Cells were exposed to DOX (0.5 μM) for 0, 0.5, 1 or 5 h, and allowed to recover for 5 or 24 h before fixation, and antibody and DAPI nuclear staining. Representative images are shown. Scale bars, 10 μm. IF images for GX or GP NS compared with WT NS are further shown in Supplementary Fig. 3b. The percentage of γH2AX-positive cells (*y* axis) at the indicated time points (*x* axis) for each group is summarized in the graph in (**e**). Error bars represent mean ± s.d. Representative 100 cells were randomly selected for quantification. Cells containing ≥ 5 foci were considered as γH2AX positive. See also Supplementary Fig. 3.

specific features driven by defective NHEJ in association with p53 deficiency.

**Attenuated PTEN DNA damage signalling promotes tumorigenicity.** To investigate whether GBM tumorigenicity necessitates

alterations in PTEN directly, we introduced lentivirus expressing the HA-tagged PTEN mutants, PTEN4E (S380E/T382E/T383E/S385E) or PTEN4A (S380A/T382A/T383A/S385A)[45], or WT PTEN cDNAs into GXP GBMs, T98G and control NS, that could be distinguished from endogenous PTEN by anti-HA

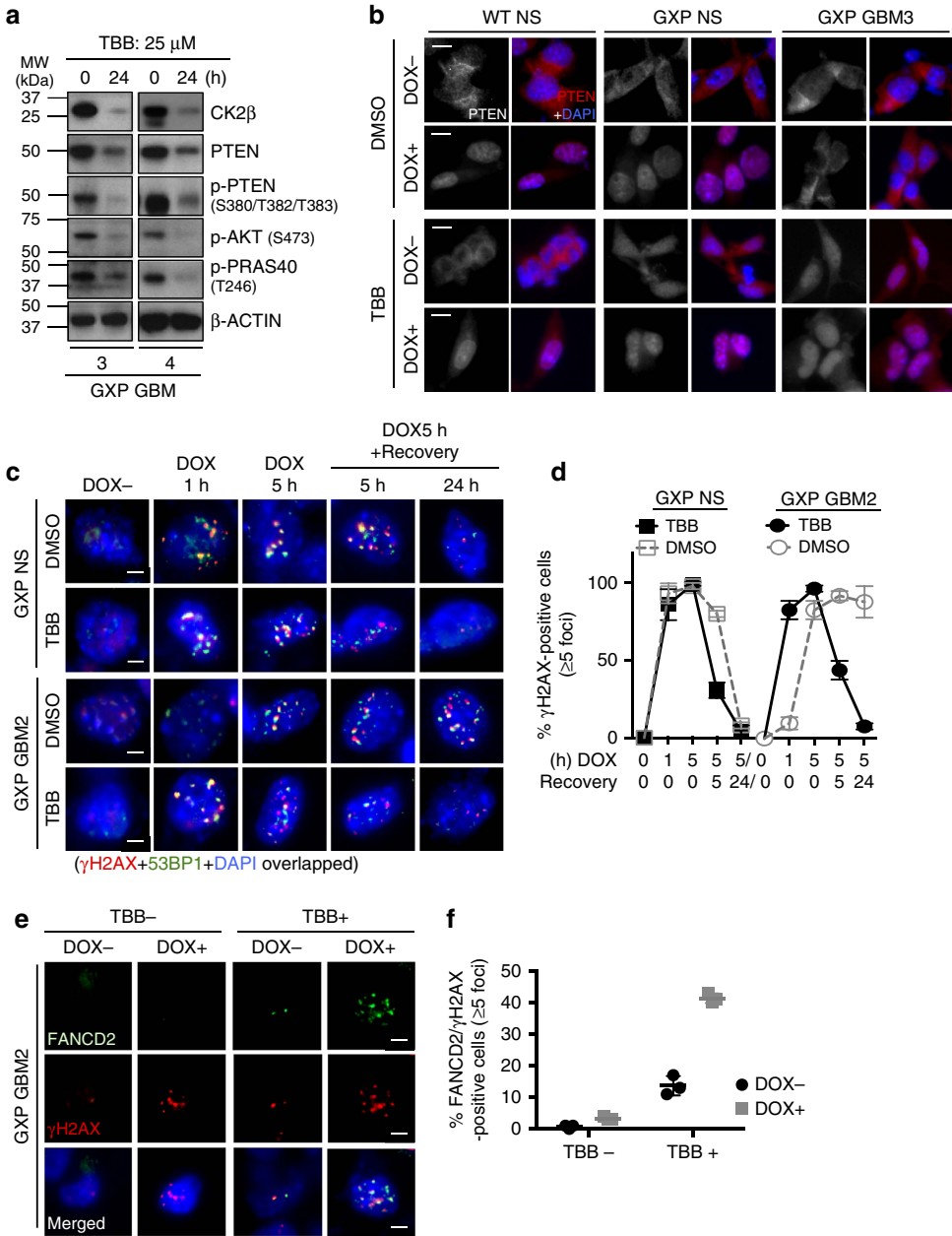

**Figure 4 | Restoration of nuclear PTEN-mediated DNA damage signalling by restraining CK2 impairs tumour but not normal neural stem cell survival.**
(**a**) Immunoblot analysis displaying the impact of TBB treatment (25 μM/24 h) on CK2β, PTEN, p-PTEN (S380/T382/T383), p-AKT (S473) and p-PRAS40 (T246) in GXP GBM3 & 4, with β-actin used as loading control. (**b**) IF of PTEN with DAPI nuclear staining in dimethylsulfoxide (DMSO)- or TBB (25 μM/24 h)-treated cells in the absence of DOX or after 5 h of exposure to DOX (0.5 μM). DOX was added 5 h before the end point of DMSO or TBB treatment. Representative images are shown. Scale bars, 20 μm. (**c**) Co-IF of γH2AX and 53BP1 foci formation with DAPI nuclear staining in DMSO- or TBB (25 μM/24 h)-treated GXP NS or GBM2 with DOX −, or 1 h or 5 h of DOX (0.5 μM) exposure followed by 5 or 24 h of recovery time. Scale bars, 10 μm. A graph quantifying the percent of γH2AX-positive GXP NS or GBM2 cells (y axis) with DMSO or TBB treatment at indicated time points of DOX treatment (x axis) is summarized in the graph in (**d**). Error bars represent mean ± s.d. Representative 100 cells were randomly selected for quantification. Cells with ≥5 foci were considered as γH2AX positive. (**e**) Co-IF of FANCD2 and γH2AX with DAPI nuclear staining in TBB (25 μM/24 h)-treated GXP GBM2 exposed to DOX (0.5 μM) 5 h before fixation. Scale bars, 10 μm. The percent of FANCD2/γH2AX-positive foci at DSB sites (y axis) in TBB − or TBB + (x axis) in GXP GBM2 is summarized in the graph shown in (**f**). Error bars represent mean ± s.e.m. See also Supplementary Fig. 4.

detection (Supplementary Fig. 6a). Like endogenous PTEN, PTEN4A (nonphosphorylatable, catalytically active) exhibited both nuclear and cytoplasmic distribution under steady-state conditions, and rapidly distributed in the nucleus in response to doxorubicin in all transduced cells (Fig. 6a,b). In contrast, PTEN4E (phosphomimetic, catalytically inactive) became nuclear in doxorubicin-treated WT and GX NS (Supplementary Fig. 6b), but remained cytoplasmic in

doxorubicin-treated GXP NS and GBMs (Fig. 6a,b) like endogenous PTEN in GXP GBMs (Fig. 3a,b and Supplementary Fig. 3a). A distinct pool of PTEN, similar to the impact of CK2β (Fig. 5f), also resided in the cytoplasm in doxorubicin-treated PTEN4E-transduced GP NS (Supplementary Fig. 6b). This suggests that aberrancies in p53 initiate mechanisms promoting PTEN loss. IF quantifying γH2AX foci further revealed that PTEN4E significantly delayed DDR kinetics in GXP NS (Fig. 6c,d),

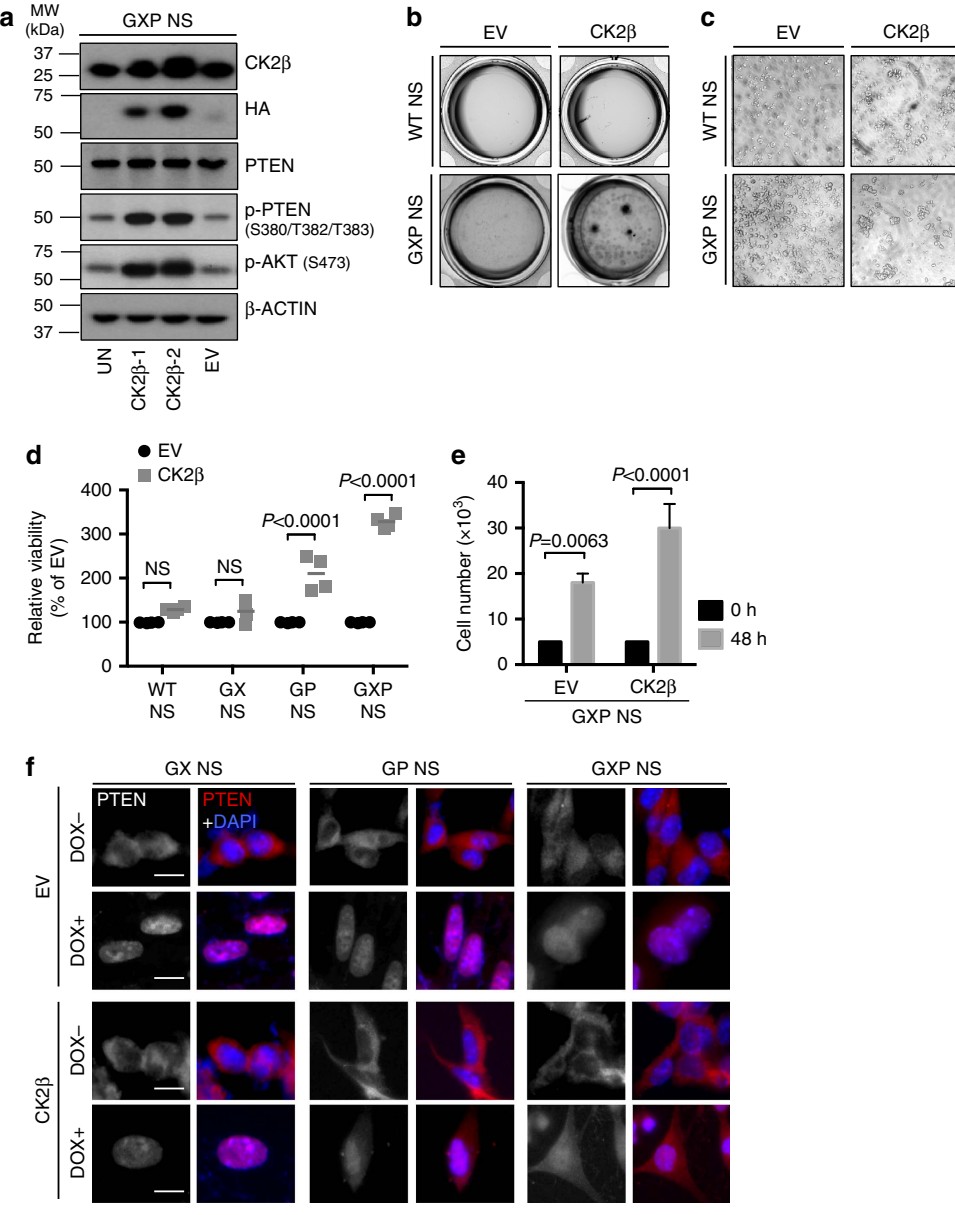

**Figure 5 | Impairment of PTEN nuclear distribution by CK2β overexpression promotes cellular transformation of GXP NS.** (**a**) Immunoblot analysis of endogenous and HA-tagged CK2β, PTEN, p-PTEN (S380/T382/T383) and p-AKT (S473) in GXP NS. (**b**–**e**) Effects of HA-CK2β cDNA overexpression on anchorage-independent growth (**b**) sphere formation in liquid cultures (**c**) and cell proliferation measured by viability (**d**) and total cell counts (**e**) assessed 48 h after seeding. Data shown for (**d**,**e**) are from three independent experiments, normalized to the value of EV-transduced cells in each group, and analysed using two-way analysis of variance (ANOVA) with Tukey's multiple comparisons test including *P* values. All graphs depict mean ± s.e.m. (**f**) IF of PTEN in CK2β-transduced GX, GP or GXP NS compared with EV-transduced GX, GP or GXP NS with or without DOX treatment (0.5 μM/5 h). Nuclei were visualized by DAPI staining. Scale bars, 20 μm. See also Supplementary Fig. 5.

comparable to the delay observed in untransduced GXP GBMs (Fig. 3d,e). Conversely, PTEN4A restored γH2AX DDR kinetics in GXP GBMs and T98G to the levels detected in untransduced GXP NS (Fig. 6c,d), effects that correlated with transcriptional activation of *PARP1*, *FANCD2*, *RAD50*, *MRE11A*, *LIG3* and *RBBP8* that were instead attenuated in PTEN4E-transduced GXP NS (Supplementary Figs 6c and 7d). Corroborating these findings, we found that formation of DNA damage-induced FANCD2 foci were reduced in PTEN4E-transduced GXP NS and GBMs (Fig. 6e) and elevated in PTEN4A-transduced GXP GBMs to the levels observed in GXP NS (Fig. 6e). Cumulatively, these data demonstrate physiological HR/A-EJ activities are suppressed in GBMs, an effect that could be mimicked by overexpressing PTEN4E or CK2β in NHEJ/p53-deficient NS. These observations

suggest that PTEN acts as an upstream sensor that sensitizes cells to DNA damage in the early phase of DDR signalling in the activation and recruitment of DDR/repair factors to DSBs. Demonstrating CK2 is limited and could be overcome by overexpressing active PTEN in GXP GBMs, we found the overexpression of WT PTEN or constitutively active PTEN4A markedly reduced the viability of GXP GBM2 but not GXP NS (Fig. 6f–h). In contrast, demonstrating PTEN inactivation induced by elevating PTEN4E (or CK2β, as above in Fig. 5) in GXP NS poised them to become malignantly transformed, overexpression of PTEN4E substantially increased GXP NS proliferative growth to levels comparable to GXP GBMs (Fig. 6f–h), moderately enhanced GX and GP NS growth and only minimally affected WT NS growth (Supplementary Fig. 6d,e).

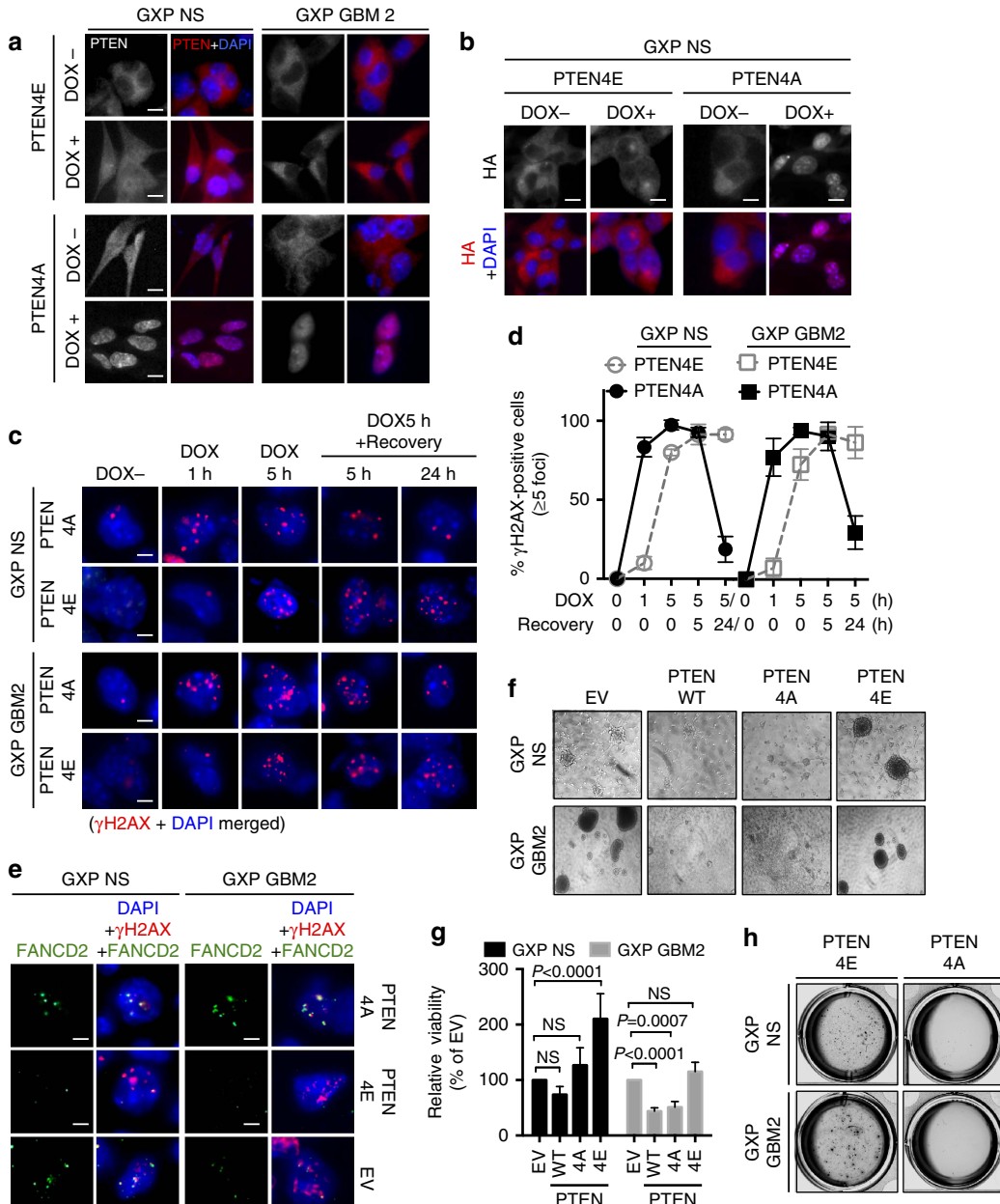

**Figure 6 | Attenuated PTEN-mediated DNA damage signalling underlies GBM tumorigenicity.** (**a**) IF of PTEN with DAPI nuclear staining in pLenti-HA-PTEN4E- or 4A-transduced GXP NS or GBM2 in the absence of DOX (dimethylsulfoxide (DMSO)-treated) or after DOX (0.5 μM/5 h) treatment. Scale bars, 10 μm. (**b**) IF of HA to localize PTEN4E or PTEN4A in pLenti-HA-PTEN4E- or 4A-transduced GXP NS with or without exposure to DOX (0.5 μM/5 h). Nuclei were visualized by DAPI staining. Scale bars, 10 μm. (**c**) IF images shown here are representatives of majority of cells in each condition. IF of γH2AX foci with DAPI staining in PTEN4A- or 4E-transduced GXP NS or GBM2: DOX − or after 1 h or 5 h of DOX (0.5 μM) exposure, followed by 5 or 24 h of recovery time. Scale bars, 10 μm. The percent of γH2AX-positive foci formed (y axis) at the indicated time points of DOX treatment (x axis) in PTEN4E- or 4A-transduced GXP NS or GBM2 is summarized in the graph shown in (**d**). Error bars represent mean ± s.d. Representative 100 cells were randomly selected for quantification and cells containing ≥5 foci were considered as γH2AX positive. (**e**) Co-IF of FANCD2 and γH2AX with DAPI nuclear staining in pLenti-HA-PTEN4A- or 4E-transduced GXP NS or GXP GBM2 after DOX (0.5 μM/5 h) treatment. Representative images are shown. Scale bars, 10 μm. (**f–h**) Sphere formation in liquid cultures (**f**) relative viability (**g**) and anchorage-independent assay (**h**) of control vector (EV), full-length WTPTEN, PTEN4A and PTEN4E-transduced GXP NS and GBM2. Normalization of data with P values from three independent experiments was performed as in Fig. 5d,e. Error bars in **g** represent mean ± s.e.m. See also Supplementary Fig. 6.

**Physiologic DNA damage signalling requires active PTEN.** Next, we transduced CK2β or PTEN4E into XRCC4$^{-/-}$/p53$^{-/-}$ mouse embryonic fibroblasts (XP$^{-/-}$ MEFs) compared with WT MEFs to ask whether changes in DDR/repair activities exerted by these proteins are generally applicable across cell types. This revealed that PTEN4E remains predominantly cytoplasmic in transduced XP$^{-/-}$ MEFs (Fig. 7a), similar to its effect in GXP NS (Fig. 6a), irrespective of DNA damage. In both CK2β- or PTEN4E-transduced XP$^{-/-}$ MEFs, the appearance and dissociation of γH2AX DNA damage-induced foci were also significantly delayed (Fig. 7b and Supplementary Fig. 7a), like in PTEN4E-transduced GXP NS (Fig. 6c,d). These data demonstrate that the pathologic impact of inactive PTEN on DDR/repair activities, exerted through CK2β or directly by PTEN4E, is not

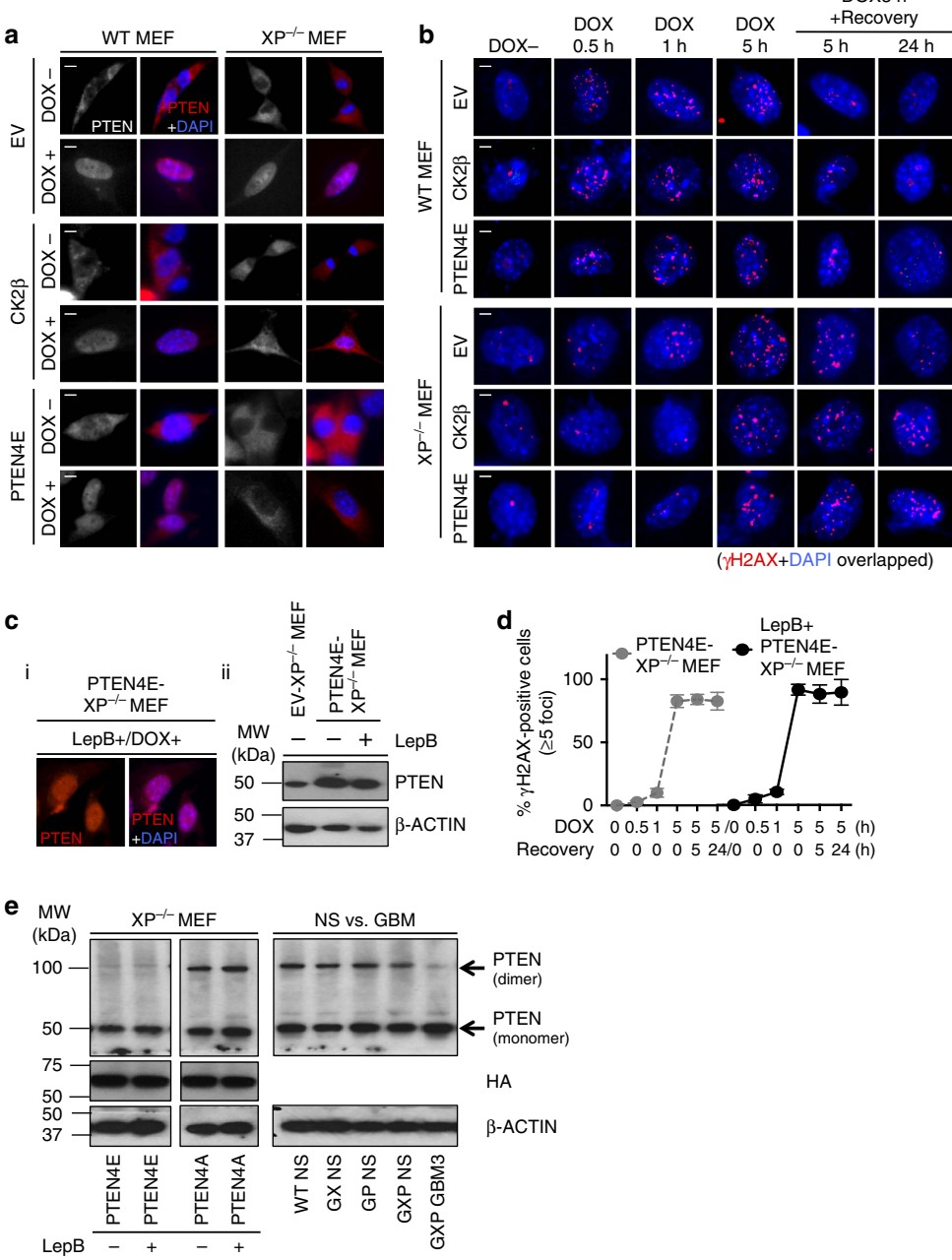

**Figure 7 | Physiologic DNA damage signalling requires nuclear distribution of active PTEN. (a)** IF of PTEN with DAPI nuclear staining in EV, CK2β or PTEN4E-transduced WT or XP$^{-/-}$ MEF cells, without DOX (dimethylsulfoxide (DMSO)-treated) or after DOX (0.5 μM/5 h) exposure. Scale bars, 10 μm. **(b)** IF of DOX-induced γH2AX foci with DAPI nuclear staining in EV, CK2β or PTEN4E-transduced WT or XP$^{-/-}$ MEF cells: DOX − or after 0.5, 1 or 5 h of DOX (0.5 μM) exposure, followed by 5 or 24 h of recovery time. Representative images are shown. Scale bars, 10 μm. **(c**i**)** IF of PTEN with DAPI nuclear staining in PTEN4E-transduced XP$^{-/-}$ MEF cells treated with Leptomycin B (LepB + ;10 ng ml$^{-1}$) 4 h before exposure to DOX. **(c**ii**)** Immunoblot analysis of PTEN in LepB-treated-PTEN4E-transduced XP$^{-/-}$ MEF cells compared with untreated PTEN4E- or EV-transduced XP$^{-/-}$ MEF cells. **(d)** A graph quantifying the percent of γH2AX-positive PTEN4E-transduced XP$^{-/-}$ MEF cells with or without LepB treatment (*y* axis) from Fig. 7b and Supplementary Fig. 7b at indicated time points of DOX treatment (*x* axis). Error bars represent mean ± s.d. Representative 50 cells were randomly selected for quantification. Cells with ≥5 foci were considered as γH2AX positive. **(e)** Immunoblot analysis of PTEN in PTEN4A- or 4E-transduced XP$^{-/-}$ MEF cells with or without LepB treatment (left panel), and in GXP GBM3 compared with WT, GX, GP or GXP NS (right panel) in nondenaturing and nonreducing conditions. See also Supplementary Fig. 7.

tissue specific, but is dependent on the combined NHEJ/p53 deficiency. Cumulatively, these results suggest that dampening of DNA damage sensors is correlated to attenuation of nuclear PTEN activity in the XP-deficient setting. This raised a key question as to whether PTEN DDR/repair activities are determined solely by its nuclear distribution or require it to be catalytically active. To test this, we treated PTEN4E-tranduced XP$^{-/}$

$^-$ MEFs with the CRM-1 nuclear export machinery inhibitor Leptomycin B[33] before exposure to doxorubicin. This revealed nuclear retention of PTEN/PTEN4E induced by Leptomycin B (Fig. 7ci) that had no obvious impact on PTEN protein levels (Fig. 7cii) and failed to restore DDR/repair activities (Fig. 7d and Supplementary Fig. 7b–d). These data definitively demonstrated that normal DDR signalling is dependent on the nuclear

distribution of active PTEN. The conformational status of inactive PTEN, based on PTEN4E, is primarily monomeric (~50 kDa), whereas catalytically active PTEN based on PTEN4A exhibits a more dimeric conformation (~100 kDa)[45]. Immunoblot analysis indeed showed that PTEN in GXP GBMs exists primarily in the monomeric state similar to PTEN4E, and in control NS, PTEN exists in both monomeric and dimeric states similar to PTEN4A (Fig. 7e). These data confirm that physiologic DDR/repair activities are blocked because of the lack of dimeric active PTEN in GXP GBMs, and imply an integral correlation between dimeric active PTEN status and its nuclear DDR/repair functions in oncogenic suppression (Fig. 8).

## Discussion

This study demonstrates that ablation of *XRCC4* in combination with p53 in neural stem/progenitors efficiently induces brain tumours at 100% penetrance, in which a heterogeneous population of NS is targeted that arise in medulloblastomas in young mice, whereas only GBMs and not other glioma subtypes

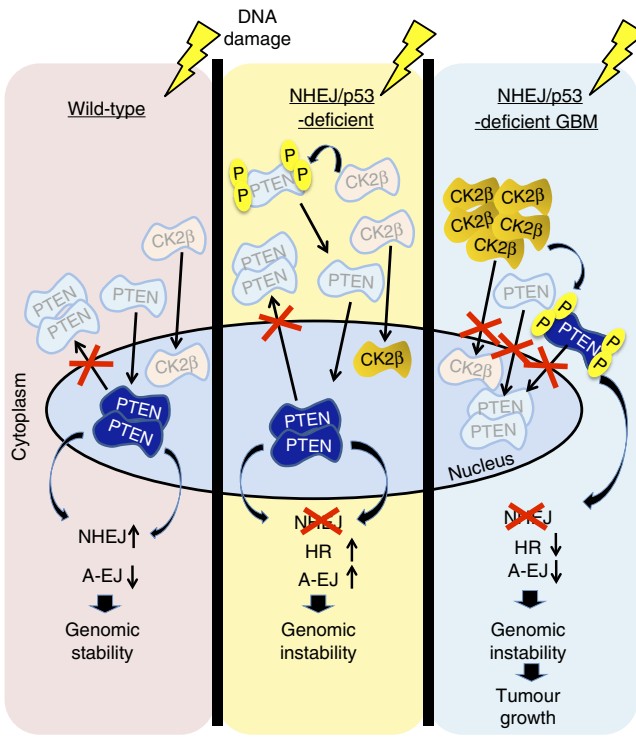

**Figure 8 | Model summarizes the contribution of defective NHEJ to PTEN DNA damage signalling and suppression of p53-mediated GBM formation and drug-resistant survival.** PTEN is a sensor that is induced as an upstream factor of DNA damage response and repair pathways. Under normal physiologic conditions, in response to DSBs, active PTEN dimers accumulate in the nucleus and activate the available DNA repair pathways, particularly NHEJ. DSBs that progress with p53 loss are accumulated further in NHEJ-deficient cells. DNA damage induced under these conditions results in constitutive activation of CK2 that is enforced in the combined NHEJ- and p53-deficient setting that inactivates and retain PTEN predominantly in its monomeric inactive state. Cells lacking active dimeric PTEN are unable to mount a DNA damage response and activate repair pathways, and constitutively activate PI3K/AKT signalling that promotes the progression of oncogenic lesions in GBM pathogenesis. Blocking the formation of the inactive p-PTEN enzyme, by restraining CK2 in the presence of DNA-damaging agents, represents a potential therapeutic strategy that suppressed resistance to genotoxic stress and GBM survival.

arise in advanced aged adult mice, with the latter more highly represented in other p53 knockout[5] and mutant models[6]. This may be because of the limited pool of adult NS in GXP (also CD21-CreX[f/f]p53[f/f]) mice that could overcome the effects of XRCC4 inactivation to oncogenically progress, whereas hGFAP-CreER or hGFAP-Cre transmission of other p53 knockout/mutant activities are implicated to target more diverse adult NS populations[6]. A surprising discovery is that generation of oncogenic alterations that impair PTEN is a key feature induced by the p53-deficient/mutant status. This was evidenced previously in p53[Ex5-6DEL] gliomas[6], and in this study, in the murine GXP and human T98G GBMs. In p53[Ex5-6DEL] mice, this is caused by significant loss of the PTEN protein[6] that, although undefined, is consistent with Notch-induced transcriptional repression and/or phosphorylated destabilization by GSK3β. In GXP GBMs and T98G, reduced *PTEN* mRNA levels are directly opposed by oncogenic CK2-induced phosphorylated-PTEN inactivation (Fig. 8). We found that CK2 is also irrevocably activated in all murine and human GBMs we tested, including the PTEN-null U87MG GBM cell line. This finding is consistent with oncogenic CK2 activity being a requisite alteration provoked in association with PTEN loss. Cumulatively, these observations are consistent with PTEN loss being integral to p53-mediated gliomagenesis, in which p53 inactivation may be a prerequisite to CK2 deregulation and inactivation of PTEN. The mechanisms induced by p53 deficiency versus mutant activities are clearly distinct. Our findings suggest that in the p53-deficient setting, XRCC4 inactivation provides a means that drives CK2 deregulation.

A key consequence of NHEJ deficiency is the accumulation of damaged DNA, principally DSBs. In B cells, we showed that DSBs in the absence of NHEJ are repaired robustly by the non-NHEJ DSB repair pathway, A-EJ[15], the key pathway that promotes the progression of oncogenic translocations in both B-cell lymphomas[18,46] and medulloblastomas[8] and also resulting from combined XRCC4 and p53 deficiencies. Although the components of the A-EJ pathway remain in contention, our studies here definitively demonstrate an integral correlation between the attenuation of non-NHEJ, HR/A-EJ genes with slower DDR/repair kinetics and DNA damage-resistant cytoplasmic PTEN retention in all the GBMs we tested. We demonstrate these effects are provoked by CK2, initiated by p53 deficiency and irrevocably deregulated by additional XRCC4 inactivation. They can be averted in GBM tumour cells by reactivating PTEN by inhibiting CK2 activity directly or transducing PTEN4A, with minimal impact on normal cells. Conversely, they can be recapitulated to induce pre/malignant phenotypes by overexpressing CK2β or PTEN4E in GXP NS and XP-deficient MEFs.

Within the nucleus, both CK2 and PTEN have purported roles in DNA repair; for CK2 in phosphorylation of DNA repair proteins/enzymes like XRCC4 and XRCC1, and for PTEN in binding to chromatin and chromatin-associated proteins involved in transcriptional regulation of DDR sensor/checkpoint and repair genes[30]. Besides CK2, PTEN nuclear activity is also presumably dependent on ATM phosphorylation or SUMOylation, as inhibition of either activity impairs PTEN nuclear expression[33]. However, the role of SUMO–PTEN remains inconclusive, as it represents a very small fraction of the total PTEN pool and exits the nucleus, whereas the predominant non-SUMO fraction is retained in the nucleus in response to DNA damage[31,33]. A possibility is that SUMO–PTEN nuclear exit may play an integral role in the early ATM-DDR signalling cascade. We demonstrate the dimeric conformational status of PTEN is essential to its nuclear activity, as Leptomycin B-induced nuclear retention of monomeric inactive PTEN4E failed to rescue

DDR/repair. Our data provide genetic evidence for a role for NHEJ gene defects in the etiology of p53-deficient GBMs, and reveal that PTEN inactivation provoked by CK2 is a critical to this malignant conversion.

Significantly, we demonstrate that the attenuation of DDR/repair induced by loss of PTEN nuclear function is a key factor that contributes to GBM genotoxic drug resistance that could be averted via methods that impair only tumours, not normal cell counterparts, for example, by reactivating PTEN by restraining CK2. We propose that the functional and molecular changes in PTEN and CK2 we uncovered could be used to increase the precision of early GBM patient stratification, and in the development of combinatorial therapies directed towards combating therapeutic resistance in GBMs with aberrancies in p53, PTEN and CK2. Future studies addressing the relevance of the cancer status of PTEN and CK2 in primary patient GBMs samples, and crosstalk and regulation of DNA damage sensing and repair pathway choice by PTEN via CK2 and/or other intersecting oncogenic pathways, may further aid to identify new therapeutic targets or methods of therapeutic intervention for this deadly disease.

## Methods

**Mouse strains and cell lines.** hGFAPCreX$^{fl/fl}$P$^{fl/fl}$, hGFAPCreX$^{fl/fl}$P$^{fl/+}$, hGFAPCreX$^{fl/fl}$ and hGFAPCreP$^{fl/fl}$ mice on a 129SvEv background were generated by breeding conditional XRCC4 (ref. 8) and p53 alleles[47] to hGFAP-Cre mice[22]. All mice were maintained in a BL1 animal facility approved by the Association for Assessment and Accreditation of Laboratory Animal Care and Institutional Animal Care and Use Committee in the Animal Research Facility at the Beth Israel Deaconess Medical Center. hGFAPCreX$^{fl/fl}$P$^{fl/fl}$ cohort mice were followed for brain tumour development that for hGFAPCreX$^{fl/fl}$P$^{fl/fl}$ mice were generally maintained up to 18 months of age. hGFAPCreX$^{fl/fl}$P$^{fl/fl}$ ($n = 37$), hGFAPCreX$^{fl/fl}$ ($n = 29$), hGFAPCreP$^{fl/fl}$ ($n = 11$) and hGFAPCreX$^{fl/fl}$P$^{fl/+}$ ($n = 7$) mice were followed for further experiments. Primers used for genotyping are detailed in the procedures of genotyping PCR assays. Human U87MG, T98G and murine GL261 cell lines were obtained from ATCC.

**Primary cell culture.** Neural stem cells obtained from P1-P3 WT (X$^{fl/fl}$P$^{fl/fl}$), GX (hGFAPCreX$^{fl/fl}$), GP (hGFAPCreP$^{fl/fl}$) or GXP (hGFAPCreX$^{fl/fl}$P$^{fl/fl}$) mouse brains, and tumour cells obtained from tumour lesions of GXP mouse brains were maintained in Dulbecco's modified Eagle's medium/F12 medium (Invitrogen) supplemented with B27 (Gibco), 40 ng ml$^{-1}$ epidermal growth factor (Peprotech), 20 ng ml$^{-1}$ basic fibroblast growth factor (Peprotech), 100 U ml$^{-1}$ penicillin, 100 mg ml$^{-1}$ streptomycin and 2 mM L-glutamine as previously described[48], and passages between 1 and 8 were used for all experiments. Cells were grown on matrigel (1:8 dilution; growth factor-reduced, BD) for further assays. WT or XP$^{-/-}$ MEF cells were isolated from E13.5 mouse embryos and maintained in Dulbecco's modified Eagle's medium (Invitrogen) supplemented with 10% fetal bovine serum, 100 U ml$^{-1}$ penicillin, 100 mg ml$^{-1}$ streptomycin and 2 mM L-glutamine as previously described[16], and passages between 1 and 5 were used for all experiments.

**Genotyping PCR assays.** PCR was performed to genotype mice and to confirm recombination and deletion at floxed alleles using genomic DNA extracted from tissues of GXP mutant mice. The list of the primer sequences used for this study is shown in Supplementary Table 3.

**Histology and immunohistochemistry.** Histology and immunohistochemistry were performed as previously described[8] using antibodies to GFAP (Cell Signaling; 3,670, 1:100), PCNA (Santa Cruz; sc-7907, 1:100) and NeuN (EMD Millipore; MAB377, 1:100) and detected with horseradish peroxidase-conjugated anti-mouse or rabbit secondary antibodies, and counterstained with 4,6-diamidino-2-phenylindole (DAPI; Sigma).

**Cytogenetic analysis of GXP GBMs.** Tumour cells were incubated in control medium with 100 nM colcemid (Karyo-Max, Invitrogen) for 3 to 5 h, dissociated and fixed as described previously[8]. Spectral karyotype analyses were performed by following standard protocols[49]. The aCGH analyses were performed for 17 labelled murine hGFAPCreX$^{fl/fl}$P$^{fl/fl}$ GBMs against matched normal tail DNA, and hybridized onto Agilent mouse 244K CGH arrays. Data were processed using Agilent software using the mouse genome data build mm9 as previously described[50].

**Analysis of differential gene expression from RNA-seq.** Differentially expressed genes were identified using Cuffdiff version 2.0.2. Paired-end RNA-seq reads were aligned to the mouse mm9 reference genome assembly, obtained from University of California at Santa Cruz (UCSC) using TopHat[51]. Quantification of reference annotations was performed to determine differential expression in known genes annotations to obtain differentially expressed genes between various WT/tumour conditions. Pathway analyses were performed using a list of 840 genes reported by The Cancer Genome Atlas (TCGA) to classify four human GBM subtypes[3,4] and a derived list of PI3K/AKT-related genes, by hierarchical clustering using R version 2.15.1 and package gplots for creating the heatmap. Log-fold changes, and in some cases raw FPKM (Fragments Per Kilobase of transcript per Million mapped reads) values, were used to generate heatmaps, with rows denoting genes and columns, and columns denoting sample conditions. The rows (genes) were scaled to have a mean of zero and a s.d. of one. The default distance function (Euclidean) was used to compute the dissimilarities between both rows and columns. The raw and normalized data have been deposited in the Gene Expression Omnibus (GEO) database (accession number: GSE75300).

**Quantitative RT–PCR-based analysis of mRNA expression.** SYBR Green (Applied Biosystem) assays were used to quantitate PI3K-related pathway or HR/A-EJ-related genes in GXP GBMs and other control samples. Total RNA at 1 μg was used to synthesize first strand of cDNA using random hexamers and SuperScript III reverse transcriptase. Primer sequence pairs used for these analyses are shown in Supplementary Table 4. Amplifications were run in a C1000 Real-time Thermal Cycler (Bio-Rad Laboratories). A DNA melt-curve was used to confirm the presence of a single PCR product in each assay. Real-time PCR results for PI3K/AKT-related genes were normalized to β-actin mRNA expression and analysed using the Mann–Whitney test and results for HR/A-EJ related gene expression were normalized and analysed using the two-way analysis of variance with Tukey's multiple comparisons test.

**IF and microscopy.** IF staining was performed as previously described[8]. DSB foci were detected by immunostaining with a monoclonal antibody to γH2AX (EMD Millipore; 05-636, 1:400), a polyclonal antibody to 53BP1 (Novus Biologicals; NB100-904, 1:100) or FANCD2 (Abcam; ab108928, 1:50). Localization studies were performed using antibody to PTEN (Cell Signalling; 9188, 1:100) or CK2β (Santa Cruz; sc-12739, 1:50) followed by incubation with a polyclonal goat anti-mouse or rabbit IgG Alexa Fluor 488/568 (Invitrogen) at 1:400 for 1 h. Cover glasses were mounted in Vectashield mountant with DAPI (Vector Laboratories) as nuclear stain. Images were captured using oil immersion 63× objectives Zeiss Axio Imager A1/Axio Cam MRC and Axiovision LE software (Carl Zeiss, Oberkochen, Germany).

**Immunoblotting and nuclear/cytoplasmic fractionation.** Immunoblotting and subcellular fractionation were performed as previously described[52]. PTEN (Cell Signalling; 9188, 1:2,000), p-PTEN (S380/T382/T383) (Cell Signalling; 9549, 1:1,000), CK2β (Santa Cruz; sc-12739, 1:100), PIK3CA (Cell Signalling; 4249, 1:1,000), PIK3CB (Santa Cruz; sc-602, 1:100), pan-AKT (Cell Signalling; 4691, 1:2,000), p-AKT (S473) (Cell Signalling; 4060, 1:1,000), p-AKT (T308) (Cell Signalling; 2965, 1:1,000), CCND1 (Cell Signalling; 2926, 1:2,000), p-PRAS40 (T246) (Cell Signalling; 2997, 1:1,000) or β-actin (Cell Signalling; 4970, 1:3,000) were used for detection followed by the incubation with peroxidase-labelled anti-mouse (Cell Signaling; 7076, 1:2,000) or anti-rabbit (BioRad; 1662408, 1:2,000) IgG secondary antibody. Image J (National Institutes of Health, Bethesda, MD, USA) was used to quantify bands and compare with the loading control. For native conditions, all reducing and denaturing reagents were excluded from all steps. Uncropped blots are shown in Supplementary Fig. 8.

**Generation of plasmid vectors.** pLenti-HA-CK2β was generated by EcoRI/BamHI insertion of mouse CK2β cDNA, and pLenti-HA-PTEN WT, PTEN4A (S380A/T382A/T383A/S385A) and PTEN4E (S380E/T382E/T383E/S385E) constructs were generated by EcoRI/BamHI insertion of the corresponding cDNA templates from pcDNA3.1 constructs (obtained from Addgene).

**Anchorage-independent assays.** $1 \times 10^5$ cells, mixed into 0.4% agar, were placed on top of a solidified medium of 0.8% noble agar, then overlaid with 500 μl of culture medium daily up to 3 weeks in 5% 37 °C $CO_2$ incubator and cells were stained with iodonitrotetrazolium chloride for colony visualization as previously described[53].

**Assessment of cell viability.** Cells were cultured in 96-well plates as $5 \times 10^3$ cells per well in control medium or with the indicated concentrations of TBB, doxorubicin, LY294002 or MK2006 or combinations. At indicated time points, quantitative determination of viability was performed with the CellTiter-Glo Luminescent Cell Viability Assay Kit according to the manufacturer's instructions (Promega). Alternatively, relative cell viability was assessed by cell counting with Trypan Blue exclusion of dead cells.

**Data availability.** RNA-seq data that support the findings of this study have been deposited in GEO with the primary accession code GSE75300. The authors declare that all other data supporting the findings of this study are available within the article and its Supplementary Information files or available from the authors on request.

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

## Acknowledgements

We thank Yan lab members for critical comments. We are indebted to Drs Pier Paolo Pandolfi, Alex Toker and Wenyi Wei for their critical and insightful discussion, and to our two DF/HCC CURE Program summer interns, Khang Nguyen and Emily Harris, for their technical support on RT–PCR validation of GXP GBM RNA-seq data. This work is supported by funding from the V Foundation, Sidney Kimmel Foundation for Cancer Research and institutional startup funds to C.T.Y.

## Author contributions

Y.-J.K., B.B. and C.T.Y. planned the studies, conducted analysis and interpretation of all experiments; Y.-J.K. and B.B. with C.T.Y. generated the GX, GP and GXP mice; Y.-J.K. conducted all experiments; E.C. performed haematoxylin & eosin (H&E) and

immunohistochemistry (IHC); R.B. analysed the primary GBMs; B.H. and H.S. contributed to analysing RNA-seq data that were validated by RT–PCR by Y.-J.K.; Y.-J.K. and C.T.Y. wrote the manuscript.

## Additional information

**Competing financial interests:** The authors declare no competing financial interests.

drug-resistant survival. *Nat. Commun.* **8,** 14013 doi: 10.1038/ncomms14013 (2017).

DOI: 10.1038/ncomms15795    **OPEN**

# Corrigendum: Contribution of classical end-joining to PTEN inactivation in p53-mediated glioblastoma formation and drug-resistant survival

Youn-Jung Kang, Barbara Balter, Eva Csizmadia, Brian Haas, Himanshu Sharma, Roderick Bronson & Catherine T. Yan

*Nature Communications* 8:14013 doi: 10.1038/ncomms14013 (2017); Published 17 Jan 2017; Updated 14 Aug 2017

'In the References section of this Article, the citation listed as reference 17 is incorrect, and should have referred to the following paper:

Forbes, S. A. *et al.* COSMIC: exploring the world's knowledge of somatic mutations in human cancer. *Nucleic Acids Res.* **43**(Database issue): D805-D811 (2015).'

