## [Peer Review File · Nature Communications]

Reviewer #1 (Remarks to the Author)

A-B. Overall, this manuscript offered insight into one major issue of GBM therapy, that being GBM's resistance to genotoxic drugs. This paper made some important findings regarding NHEJ, PTEN, and p53 in GBM with potential clinical application. In generating a XRCC4/P53 deleted mouse model driven by hGFAP-Cre, it was found that high-grade astrocytomas, specifically GBMs, develop among aged animals. Among these tumors, PTEN mRNA was reduced and the protein was aberrantly localized which was found to be dependent on its phosphorylation status. This was resultant of elevated CK2 expression in these tumors. Reactivating PTEN in vitro (relocalizing it to the nucleus) via inhibiting CK2, it was shown that genotoxic drug resistance can be obviated, thereby showing the therapeutic potential of a CK2 inhibitor that could target GBM cells that are resistant to DNA damage inducing drugs. Additionally, several PTEN mutants were generated to show that PTEN acts as a sensor to sensitize cells to DNA damage and that it is the nuclear localization of PTEN that promotes its DNA damage response activities. Lastly, it was shown that PTEN localization and subsequent DNA damage responsiveness is highly dependent on p53 status. This manuscript does offer insight into mechanisms of GBM resistance, a problem that plagues current therapies.

C-E. Data was well-interpreted and presented with proper use of statistics. Additionally, several models and lines were used which lends credence to the conclusions that were drawn. It is recommended to clarify Figure 1E as it is unclear how the conclusion was made that the GBMs that were generated in the mice fell into the Proneural or Classical subgroup of GBMs. This was not explained in the Methods section and it is unclear how the conclusion was made.

F. As mentioned, overall, the data is sound and convincing. My recommendation to accept with revisions is simply to go over the contents of the manuscript and revise several portions which are rather unclear, and at times, contradicting. Specifically:
Lines 101-102 - sentences contradict each other. First sentence says none of the cohorts developed tumors. The following sentence states that mice developed brain tumors with 100% penetrance.
Lines 115-117 - confusing, especially because CD21-Cre is mentioned. It is understood that this is from the groups previous publication, but bringing this into the picture at this portion of the paragraph makes it difficult to understand. I think the attempt that would ultimately lead to a good point, but as written, it is confusing. A more direct statement would be better received/interpreted by the reader.
Lines 143-145 - very confusing as written.
Lines 264-265 - confusing.
Lines 334-337 - XRCC4 and P53 deletion induced GBM as well as MB. Granted, among the "aged adult" animals, only GBM developed, but this sentence is a bit misleading. It is interesting, though not fully addressed that you can get histologically different tumors using the same mutations. This likely has to do with the broadly expressed hGFAP-Cre and the different cells that are inevitably deleting XRCC4 and P53, therefore, potentially difference cells of origin are being targeted.
Line 352-354-unclear what is meant.
Line 427-428 - distinction between row and column (rows are horizontal, columns are vertical).

The above changes/revisions would be highly recommended prior to publication.

The following comment would make for a more therapeutically relevant piece of work, though not necessary for this particular manuscript. Being the in vivo models have been established, and the in vitro data was encouraging, this paper could have a stronger impact if an additional experiment was performed utilizing the inhibitor on the mouse models and showing therapeutic efficacy.

G-H. Content of the abstract, introduction, and conclusions were fair and references were appropriate to form the foundation of the manuscript and corroborate the presented data. As mentioned, it would be strongly recommended to revise several portions (mentioned above) to ensure a proper flow and understanding of the material. As is currently written, the writing is not

as clear as it could be and therefore causes the reader confusion. This can be easily rectified with the suggested editorial revisions and with the strong data, should not cause issue.

Reviewer #2 (Remarks to the Author)

Kang et al. generated a mouse model for glioblastoma multiforme (GBM) by conditional knock-out of XRCC4 and p53. These mice develop tumors that exhibit reduced PTEN function due to reduced PTEN mRNA and increased CK2-mediated phosphorylation and inactivation of PTEN. PTEN and CK2 remain localized in the cytoplasm in GXP transgenic GBM tumor cells, in contrast to nuclear translocation in response to DNA damage in neural stem cells, which is accompanied by an impaired DNA damage response in GXP-GBM cells. TBB-mediated inhibition of CK2 function reduced PTEN function and restored the DNA damage response in GXP-GBM cells. Overexpression of CK2 in GXP neural stem cells impaired nuclear localization of PTEN and promoted transformation. Expression of non-phosphorylatable, active PTEN (PTEN4A), but not phosphomimetic, inactive PTEN4E, restored nuclear localization of PTEN and rescued the DNA damage response in GXP-GBM2 cells. The DNA damage response in XP^{-/-} MEF cells was impaired by expression of CK2 or PTEN4E. Nuclear retention of inactive PTEN4E failed to restore the DNA damage response. PTEN dimerized to a lesser extent in GXP-GBM3 cells than in neural stem cells, suggesting that active PTEN dimers are required for the DNA damage response. The authors conclude that "these observations definitively demonstrate NHEJ contributes to p53-mediated glioblastoma suppression, and reveal a crucial role for PTEN in the early DNA damage-signaling cascade, the inhibition of which promotes glioma tumorigenicity and drug-resistant survival."

The authors developed an interesting model for GBM and demonstrate that aberrant regulation of PTEN is involved in tumor development. Whereas the main findings are interesting, the data are overinterpreted and/or misrepresented in the text in multiple instances as outlined below, which hampers proper assessment of whether the conclusions are supported by the data.

Points:

1. Fig. 2. PTEN RNA levels are downregulated in GXP GBMs (Fig. 2a), but PTEN protein levels appear not to be affected (Fig. 2d). From the overexposed immunoblot in Fig. 2d, it is hard to judge whether there are differences in PTEN protein levels, although it appears there are no gross changes in PTEN protein. The pPTEN blot is also overexposed and from the blot it is not evident that PTEN phosphorylation is enhanced in GXP GBMs compared to GXP NS. CK2beta protein levels appear to be enhanced in GXP GBMs, albeit only a modest increase was observed compared to GXP NS. The results in Fig. 2b (which are not described in the text) show that pAKT levels appear not to be correlated to PIK3CA or PIK3CB levels, or to PTEN levels (Fig. 2d), suggesting another mechanism underlies AKT activation. The discrepancies in these data should be explained.
2. Fig. 4. The competitive CK2 inhibitor, TBB, reduced CK2 and PTEN protein levels and reduced PTEN phosphorylation, resulting in reduced pAKT and pPRAS40 levels. Despite the clear reduction in PTEN level, downstream AKT signaling is reduced, which would be consistent with enhanced PTEN activity. Yet, PTEN protein is predominantly localized in the nucleus in GXP GBM3 cells (Fig. 4b), which is not consistent with reduced pAKT levels. This should be explained. Quantification of the nuclear and cytoplasmic PTEN by cell fractionation as was done in Fig. 3b, would help to quantify the effect of TBB on subcellular localization of PTEN.
3. Fig. 5. CK2beta expression in GXP NS cells was modest (Fig. 5a), but had profound effects on cell proliferation (Fig. 5b-e). The authors claim (p.12) that CK2beta expression in combination with doxorubicin resulted in substantial loss of nuclear expression of PTEN in GXP NS cells. However, this is not evident from Fig. 5f (bottom right panel).
4. Fig. 6a, 7a, p.14. "PTEN4E remains exclusively cytoplasmic in transduced XP^{-/-} MEFs (Fig. 7a), similar to its effect in GXP NS (Fig. 6a)." This is an overstatement of the results. Doxorubicin treatment of GXP NS cells does not result in exclusive cytoplasmic localization of PTEN4E (Fig. 6a).
5. Fig. 7c II. pPTEN levels are upregulated in response to expression of PTEN4E. What is the origin of the enhanced signal? Presumably, PTEN4E itself is not recognized by the pPTEN-specific

antibody.

Minor point

1. For clarity, the title of the section "Overexpressing CK2beta by impairing PTEN transforms XRCC4/p53 deficient cells" should be rephrased to: "Overexpressing CK2beta transforms XRCC4/p53 deficient cells by impairing PTEN".

Reviewer #3 (Remarks to the Author)

In this manuscript Yan and colleagues use mouse models to investigate the role of p53 and the DNA damage response in GBM. The manuscript is very well written and represents a great deal of work. The authors find that depletion of p53 in XRCC4 deficient mice leads to high penetrance brain tumours and reduction of PTEN mRNA. They propose that DNA damage induced by XRCC4/p53 deletion causes elevated CK2 dependent phosphorylation of PTEN that regulates its intracellular activities. These studies add to a body of evidence linking PTEN with the DNA damage response, and provide clues to possible mechanisms of GBM initiation in patients.

Minor comments:

Figure 1b. It would aid the reader if the abbreviations GXP etc as defined in the text could be added to the figure along with hGFAPCre... etc

Figure 2b: what do the red arrows represent

Scale bars should be added to IF images

The quality of the blots in Figure 3b are rather poor. Can they be improved? Are they representative of multiple experiments?

Response to Reviewer #1:

A-B. Overall, this manuscript offered insight into one major issue of GBM therapy, that being GBM's resistance to genotoxic drugs. This paper made some important findings regarding NHEJ, PTEN, and p53 in GBM with potential clinical application. In generating a XRCC4/P53 deleted mouse model driven by hGFAP-Cre, it was found that high-grade astrocytomas, specifically GBMs, develop among aged animals. Among these tumors, PTEN mRNA was reduced and the protein was aberrantly localized which was found to be dependent on its phosphorylation status. This was resultant of elevated CK2 expression in these tumors. Reactivating PTEN in vitro (relocalizing it to the nucleus) via inhibiting CK2, it was shown that genotoxic drug resistance can be obviated, thereby showing the therapeutic potential of a CK2 inhibitor that could target GBM cells that are resistant to DNA damage inducing drugs. Additionally, several PTEN mutants were generated to show that PTEN acts as a sensor to sensitize cells to DNA damage and that it is the nuclear localization of PTEN that promotes its DNA damage response activities. Lastly, it was shown that PTEN localization and subsequent DNA damage responsiveness is highly dependent on p53 status. This manuscript does offer insight into mechanisms of GBM resistance, a problem that plagues current therapies.

C-E. Data was well-interpreted and presented with proper use of statistics. Additionally, several models and lines were used which lends credence to the conclusions that were drawn.

-Our thanks to reviewer #1 for his/her positive comments and helpful suggestions. We have corrected all sentences, which were pointed as unclear and confusing. We hope the changes made provide sufficient clarification and are considered acceptable.

It is recommended to clarify...

Point-by-point reply to specific comments:

1. Figure 1E as it is unclear how the conclusion was made that the GBMs that were generated in the mice fell into the Proneural or Classical subgroup of GBMs. This was not explained in the Methods section and it is unclear how the conclusion was made.

-This conclusion is generated based on our expression analyses, comparing the GBM classification from TCGA (ref. 3 and 4) to our RNA-seq data for 6 independent GXP GBMs, and which we subsequently validated in the remaining 11 GXP GBMs using RT-PCR. This explanation, which we hope better clarifies how this conclusion was made, is now more carefully detailed in the highlighted revised manuscript, in the main text (lines; 130-133), the Methods section (lines; 451-454), and within the Figure legend for Fig. 1e (lines; 728-734).

F. As mentioned, overall, the data is sound and convincing. My recommendation to accept with revisions is simply to go over the contents of the manuscript and revise several portions which are rather unclear, and at times, contradicting.

-We thank reviewer #1 for his/her recommended revision in which the writing was unclear or considered contradicting. We have made every attempt to clarify all these portions, which we have detailed below and highlighted in the revised manuscript.

Specifically:

2. Lines 101-102- sentences contradict each other. First sentence says none of the cohorts developed tumors. The following sentence states that mice developed brain tumors with 100% penetrance.

-We should have stated more clearly, that none of the GX, GP, and GXP^{het} cohorts developed tumours, while all the experimental GXP mice developed brain tumours with 100% penetrance. The specific cohorts are now added and highlighted in the revised text as follows:

From: “Significantly, while none of the cohort mice developed tumours (Fig.1b), GXP mice ($n=37$) developed brain tumours with 100% penetrance.”

To “Significantly, while none of the cohort GX, GP, and GXP^{het} mice developed tumours (Fig. 1b), GXP mice ($n=37$) developed brain tumours with 100% penetrance.” (Lines; 102-104 in the revised manuscript)

3. Lines 115-117- confusing, especially because CD21-Cre is mentioned. It is understood that this is from the groups previous publication, but bringing this into the picture at this portion of the paragraph makes it difficult to understand. I think the attempt that would ultimately lead to a good point, but as written, it is confusing. A more direct statement would be better received/interpreted by the reader.

-To clarify further the statement has been modified as follows:

From “These comparisons suggest GBM precursors, which based on CD21-Cre expression may originate from SGZ NS, may be particularly vulnerable to simultaneous ablation of *XRCC4* and p53.”

To “In adult GXP (also CD21-CreX^{ff}p53^{ff} 18) mice, it may be that simultaneous ablation of *XRCC4* and p53 provides selective growth advantage to very limited pool of NS during adult neurogenesis, which based on the intersection of adult NS that express both hGFAP-Cre and CD21-Cre²⁰, may originate from the adult SGZ.” (Lines; 117-120 in the revised manuscript)

4. Lines 143-145 - very confusing as written. “To determine if concordant with the reduction in PTEN mRNA, PTEN protein levels are correspondingly reduced or lost in the GBMs induced in GXP mice, similar to 80% of p53^{Ex5-6DEL} gliomas⁶, we performed anti-PTEN immunofluorescence (IF) staining on paraffin-embedded primary GBM-bearing brain sections.”

-To better clarify, this text has been modified to:

“To determine if PTEN protein levels are concordant with the reduction in *PTEN* mRNA in GXP GBMs, similar to the observation in 80% of p53^{Ex5-6DEL} gliomas⁶, we performed anti-PTEN immunofluorescence (IF) staining on paraffin-embedded primary GBM-bearing brain sections.” (Lines; 152-155 in the revised manuscript)

5. Lines 264-265 - confusing.

“To investigate if GBM tumorigenicity necessitates alterations in PTEN directly, or is indirectly exerted by inactivation of PTEN provoked by CK2, we introduced lentivirus.”

-To better clarify, the text has been modified to:

“To investigate if GBM tumorigenicity necessitates alterations in PTEN directly, we introduced lentivirus...” (Lines; 277-278 in the revised manuscript)

6. Lines 334-337 - XRCC4 and P53 deletion induced GBM as well as MB. Granted, among the "aged adult" animals, only GBM developed, but this sentence is a bit misleading. It is interesting, though not fully addressed that you can get histologically different tumors using the same mutations. This likely has to do with the broadly expressed hGFAP-Cre and the different cells that are inevitably deleting XRCC4 and p53, therefore, potentially difference cells of origin are being targeted.

-We thank the reviewer for this very insightful comment. In the context of medulloblastomas, indeed our cytogenetic analysis revealed more heterogeneous chromosomal abnormalities than those we obtained previously by utilising Nestin-Cre (ref. 8). As suggested by this reviewer, this likely has to do with the broad expression of hGFAP-Cre and simultaneous ablation of *XRCC4* and p53 in different neural stem/progenitor cell populations and therefore, potential differences in the cells of origin are being targeted. In adult GXP (also CD21-CreX^{ff}p53^{ff}; ref. 18) mice, it appears that a very limited pool of NS that could overcome the effects of *XRCC4* inactivation to oncogenically progress is targeted, whereas hGFAP-CreER or hGFAP-Cre transmission of other p53 knockout/mutant activities are implicated to target more diverse adult NS populations. Accordingly, we have modified this sentence to better discuss tumour cell of origin, which we have detailed below and highlighted in the revised manuscript.

The text has been modified:

From “ This study demonstrates that ablation of *XRCC4* in combination with p53 in neural stem/progenitors efficiently induces GBMs and not other glioma subtypes in advanced aged adult mice, with the latter more highly represented in other p53 knockout⁵ and mutant models⁶.”

To “This study demonstrates that ablation of *XRCC4* in combination with p53 in neural stem/progenitors efficiently induces brain tumours at 100% penetrance, in which a heterogeneous population of NS is targeted that arise in medulloblastomas in young mice, while only GBMs and not other glioma subtypes in advanced aged adult mice, with the latter more highly represented in other p53 knockout⁵ and mutant models⁶.” (Lines; 345-349 in the revised manuscript)

7. Line 352-354-unclear what is meant. This finding is consistent with oncogenic CK2 activity being a requisite alteration provoked in association with PTEN loss. “Cumulatively, these observations are consistent with PTEN loss being integral to p53-mediated gliomagenesis, although the mechanisms induced by p53-deficiency versus mutant activities are clearly distinct.”

-We mean the initiating events promoting GBMs may be stepwise. p53-deficiency is a pre-requisite to CK2 deregulation and inactivation of PTEN, where CK2 deregulation is instigated by *XRCC4* inactivation in our model. To better clarify what we mean, the text has been modified and highlighted in the revised manuscript, which we have detailed below:

From “Cumulatively, these observations are consistent with PTEN loss being integral to p53-mediated gliomagenesis, although the mechanisms induced by p53-deficiency versus mutant activities are clearly distinct.”

To “Cumulatively, these observations are consistent with PTEN loss being integral to p53-mediated gliomagenesis, in which p53 inactivation may be a pre-requisite to CK2 deregulation and inactivation of PTEN. The mechanisms induced by p53-deficiency versus mutant activities are clearly distinct, which in our model in the p53-deficient setting, our findings suggest CK2 deregulation is instigated by *XRCC4* inactivation.” (Lines; 364-368 in the revised manuscript)

8. Line 427-428 - distinction between row and column (rows are horizontal, columns are vertical).

-We thank the reviewer for this correction and have modified the text:

From: “Each column represents a distinct sample (6 independent GXP GBMs compared to GXP NS) and each row represents an individual gene”

To: “Each row represents a distinct sample (6 independent GXP GBMs compared to GXP NS) and each column represents an individual gene...” (Lines; 729-730 in the revised manuscript)

The above changes/revisions would be highly recommended prior to publication.

The following comment would make for a more therapeutically relevant piece of work, though not necessary for this particular manuscript. Being the in vivo models have been established, and the in vitro data was encouraging, this paper could have a stronger impact if an additional experiment was performed utilizing the inhibitor on the mouse models and showing therapeutic efficacy.

-As suggested by this reviewer, for our future study and beyond the scope of this current manuscript, we are indeed performing additional experiments testing the therapeutic efficacy of the CK2 inhibitor/doxorubicin combination.

G-H. Content of the abstract, introduction, and conclusions were fair and references were appropriate to form the foundation of the manuscript and corroborate the presented data. As mentioned, it would be strongly recommended to revise several portions (mentioned above) to ensure a proper flow and understanding of the material. As is currently written, the writing is not as clear as it could be and therefore causes the reader confusion. This can be easily rectified with the suggested editorial revisions and with the strong data, should not cause issue.

-We have made every effort to make all changes/revisions recommended above for publication. We thank

reviewer #1 for his/her careful perusal and strong support of our work.

Response to Reviewer #2:

Kang et al. generated a mouse model for glioblastoma multiforme (GBM) by conditional knock-out of XRCC4 and p53. These mice develop tumors that exhibit reduced PTEN function due to reduced PTEN mRNA and increased CK2-mediated phosphorylation and inactivation of PTEN. PTEN and CK2 remain localized in the cytoplasm in GXP transgenic GBM tumor cells, in contrast to nuclear translocation in response to DNA damage in neural stem cells, which is accompanied by an impaired DNA damage response in GXP-GBM cells. TBB-mediated inhibition of CK2 function reduced PTEN function and restored the DNA damage response in GXP-GBM cells. Overexpression of CK2 in GXP neural stem cells impaired nuclear localization of PTEN and promoted transformation. Expression of non-phosphorylatable, active PTEN (PTEN4A), but not phosphomimetic, inactive PTEN4E, restored nuclear localization of PTEN and rescued the DNA damage response in GXP-GBM2 cells. The DNA damage response in XP^{-/-} MEF cells was impaired by expression of CK2 or PTEN4E. Nuclear retention of inactive PTEN4E failed to restore the DNA damage response. PTEN dimerized to a lesser extent in GXP-GBM3 cells than in neural stem cells, suggesting that active PTEN dimers are required for the DNA damage response. The authors conclude that "these observations definitively demonstrate NHEJ contributes to p53-mediated glioblastoma suppression, and reveal a crucial role for PTEN in the early DNA damage-signaling cascade, the inhibition of which promotes glioma tumorigenicity and drug-resistant survival."

The authors developed an interesting model for GBM and demonstrate that aberrant regulation of PTEN is involved in tumor development. Whereas the main findings are interesting, the data are overinterpreted and/or misrepresented in the text in multiple instances as outlined below, which hampers proper assessment of whether the conclusions are supported by the data.

-We thank reviewer #2 for his/her critical review of our overall data. We have made every effort to address all the comments detailed below, to ensure clarity and as further support our data, which we hope will be considered satisfactory.

Point-by-point reply to specific comments:

1-1. Fig. 2. PTEN RNA levels are downregulated in GXP GBMs (Fig. 2a), but PTEN protein levels appear not to be affected (Fig. 2d). From the overexposed immunoblot in Fig. 2d, it is hard to judge whether there are differences in PTEN protein levels, although it appears there are no gross changes in PTEN protein. The pPTEN blot is also overexposed and from the blot it is not evident that PTEN phosphorylation is enhanced in GXP GBMs compared to GXP NS. CK2beta protein levels appear to be enhanced in GXP GBMs, albeit only a modest increase was observed compared to GXP NS.

-We have re-probed the PTEN and p-PTEN immunoblots for Fig. 2d to show the lower exposures that will allow the reviewer to better judge whether there are any modest differences in PTEN protein levels and increase in p-PTEN levels in GXP GBMs.

1-2. The results in Fig. 2b (which are not described in the text) show that p-AKT levels appear not to be correlated to PIK3CA or PIK3CB levels, or to PTEN levels (Fig. 2d), suggesting another mechanism underlies AKT activation. The discrepancies in these data should be explained.

-We thank the reviewer for noting the omission of text describing Fig. 2b, and also his/her suggestion of the possibility of another mechanisms underlying AKT activation. In the context, our data shows the levels of PIK3CB and p-AKT (S473) levels to correlate directly, while the levels of PIK3CA and p-AKT (T308) appeared to be more heterogeneous. This explanation, as well as text describing Fig 2b, is now provided and highlighted in our revised text as follows:

“To validate the induction of PI3K/AKT pathway proteins, we performed immunoblot analysis, examining the expression levels of PIK3CA, PIK3CB and p-AKT in 7 representative independent GXP GBMs. This revealed the levels of PIK3CB and p-AKT (S473) were elevated in all 7 GXP GBMs, and in all the GBM cell lines we tested compared to WT NS or WT brain, while the levels of PIK3CA and p-AKT (T308) appeared to be more heterogeneously expressed (Fig. 2b).” (Lines; 140-145 in the revised manuscript)

2-1. Fig. 4. The competitive CK2 inhibitor, TBB, reduced CK2 and PTEN protein levels and reduced PTEN phosphorylation, resulting in reduced p-AKT and pPRAS40 levels. Despite the clear reduction in PTEN level, downstream AKT signaling is reduced, which would be consistent with enhanced PTEN activity. Yet, PTEN protein is predominantly localized in the nucleus in GXP GBM3 cells (Fig. 4b), which is not consistent with reduced p-AKT levels. This should be explained.

-We know that in the GBMs in our model, *PTEN* mRNA levels are low but the PTEN protein levels are comparable to neural stem cells, because of phosphorylation by CK2. TBB, by inhibiting CK2, results in the dephosphorylation and destabilisation of PTEN, which while transcriptionally reduced, is now catalytically active, able to localise to the nucleus in response to DNA damage and block PI3K/AKT signaling. To better clarify our findings, we have modified the text within lines; 224-228 to:

“Therefore, to test if pathological CK2 activity and PTEN subcellular distribution are coordinately regulated, we treated GXP GBMs, T98G and control NS with TBB, which by competitively inhibiting CK2 kinase activity, should reduce CK2-mediated PTEN phosphorylation and restore PTEN’s protein turnover rate (Supplementary Fig. 2e,f) to reflect its physiologic reduced mRNA levels in GXP GBMs (Supplementary Fig. 2a).” (Underlined text here describes the key changes in the text in this section).

Explanation for the reduction in p-AKT levels is detailed within highlighted text within lines; 229-236, which we underline and detail below:

“Indeed, immunoblot analysis revealed that TBB reduced CK2, total PTEN and p-PTEN protein levels, and despite its reduced expression, restored PTEN’s catalytic activity and ability to negatively regulate PI3K/AKT signaling, the latter shown by the reduction in p-AKT and p-PRAS40 in GXP GBMs (Fig. 4a).”

2-2. Quantification of the nuclear and cytoplasmic PTEN by cell fractionation as was done in Fig. 3b, would help to quantify the effect of TBB on subcellular localization of PTEN.

- As suggested, we further used cell fractionation to quantify PTEN nuclear and cytoplasmic distribution in GXP GBMs in response to TBB treatment. The pattern of PTEN distribution observed in immunoblot analysis of nuclear/cytoplasmic fractionation is consistent with IF data for Fig. 4b. Immunoblots for fractionation data are now shown in Supplementary Fig. 4b and described and highlighted in the revised manuscript within lines; 232-236 in the main text and lines; 75-77 in the Supplementary information.

We note that in order to place these data in Supplementary Fig. 4b, Supplementary Figures (Supplementary Fig. 4, 5, and 6) were shifted and re-numbered.

3-1. Fig. 5. CK2beta expression in GXP NS cells was modest (Fig. 5a), but had profound effects on cell proliferation (Fig. 5b-e).

-Data in Fig. 5 describe the effects of CK2 β overexpression, not endogenous CK2, and show that the overexpression of CK2 β profoundly increases the proliferation of GXP NS (XRCC4/p53-deficient neural stem cells), comparable to the levels observed for GXP GBMs.

3-2. The authors claim (p.12) that CK2beta expression in combination with doxorubicin resulted in substantial loss of nuclear expression of PTEN in GXP NS cells. However, this is not evident from Fig. 5f (bottom right panel).

- In the top panels in Fig. 5f: With transduction of empty vector (EV), PTEN is cytoplasmic in GX, GP and GXP NS under steady-state conditions, and becomes nuclear in response to DNA damage.
- In the bottom panels of Fig. 5f, with lentiviral transduction of CK2 cDNA: In the absence of doxorubicin (DOX-): Immunofluorescence shows PTEN is predominantly cytoplasmic in GX, GP and GXP NS. Then with addition of doxorubicin (DOX+): Left panel shows a significant

proportion of cytoplasmic PTEN remains in GP NS (middle panel) and GXP NS (right panel), whereas DNA damage induced in CK2 β -transduced GX NS resulted in a normal pattern of DNA damage induced PTEN nuclear expression.

Explanation better clarifying this data has been modified within lines; 267-270 in the highlighted text in the manuscript, and is underlined below:

“However, whereas PTEN is predominantly redistributed to the nucleus in GX, GP and GXP NS in response to doxorubicin-induced DNA damage, a substantial fraction of PTEN remained cytoplasmic in doxorubicin treated CK2 β -transduced GP and GXP NS (Fig. 5f).”

4. Fig. 6a, 7a, p.14. "PTEN4E remains exclusively cytoplasmic in transduced XP^{-/-} MEFs (Fig. 7a), similar to its effect in GXP NS (Fig. 6a)." This is an overstatement of the results. Doxorubicin treatment of GXP NS cells does not result in exclusive cytoplasmic localization of PTEN4E (Fig. 6a).

-The text has been modified to better clarify as follows:

From "PTEN4E remains exclusively cytoplasmic in transduced XP^{-/-} MEFs (Fig. 7a), similar to its effect in GXP NS (Fig. 6a)."

To “This revealed that PTEN4E remains predominantly cytoplasmic in transduced XP^{-/-} MEFs (Fig. 7a), similar to its effect in GXP NS (Fig. 6a)” (Lines; 317-319 in the revised manuscript).

5. Fig. 7c II. pPTEN levels are upregulated in response to expression of PTEN4E. What is the origin of the enhanced signal? Presumably, PTEN4E itself is not recognized by the pPTEN-specific antibody.

-This row should have actually been labeled as PTEN instead of p-PTEN. We apologise for the mislabeling and thank the reviewer for finding the mistake. This has been corrected to PTEN and the change in text is highlighted in the revised manuscript (Lines; 331-332 in the revised manuscript).

Minor point

1. For clarity, the title of the section "Overexpressing CK2beta by impairing PTEN transforms XRCC4/p53 deficient cells" should be rephrased to: "Overexpressing CK2beta transforms XRCC4/p53 deficient cells by impairing PTEN".

-We thank the reviewer for correcting this sentence, which we have now corrected as suggested. (Line; 260 in the revised manuscript)

Reviewer #3 (Remarks to the Author):

In this manuscript Yan and colleagues use mouse models to investigate the role of p53 and the DNA damage response in GBM. The manuscript is very well written and represents a great deal of work. The authors find that depletion of p53 in XRCC4 deficient mice leads to high penetrance brain tumours and reduction of PTEN mRNA. They propose that DNA damage induced by XRCC4/p53 deletion causes elevated CK2 dependent phosphorylation of PTEN that regulates its intracellular activities. These studies add to a body of evidence linking PTEN with the DNA damage response, and provide clues to possible mechanisms of GBM initiation in patients.

-We thank reviewer #3 for their positive comments, and particularly, their suggestion for our overall data. We have made every effort to address all the comments to ensure clarity, which we hope will satisfy this reviewer.

Point-by-point reply to specific comments:

Minor comments:

1. Figure 1b. It would aid the reader if the abbreviations GXP etc as defined in the text could be added to the figure along with hGFAPCre... etc

-We thank the reviewer for suggesting this point, which we have now corrected in Fig. 1b to aid further clarification for readers as follows:

From: hGFAPCreX^{fl/fl}P^{fl/fl}-GBM

hGFAPCreX^{fl/fl}P^{fl/fl}-MB
hGFAPCreX^{fl/fl}
hGFAPCreP^{fl/fl}
hGFAPCreX^{fl/fl}P^{fl/+}

To: hGFAPCreX^{fl/fl}P^{fl/fl}-GBM;GXP GBM

hGFAPCreX^{fl/fl}P^{fl/fl}-MB;GXP MB
hGFAPCreX^{fl/fl};GX
hGFAPCreP^{fl/fl};GP
hGFAPCreX^{fl/fl}P^{fl/+};GXP^{het}

2. Figure 2b: what do the red arrows represent

-The red arrows in Fig. 2b indicate bands for PIK3CB and p-AKT (T308), respectively, which we now better clarify in the legend for this figure. The change in text is detailed below and highlighted in the revised manuscript.

(b) Immunoblot analysis of PIK3CA, PIK3CB (indicated by red arrow), pan-AKT, p-AKT (S473), p-AKT (T308) (indicated by red arrow), and CCND1 in 7 independent GXP GBMs compared to WT or GXP NS, WT adult forebrain, GL261 (p53-mutated), U87MG (PTEN-null), or T98G (p53/PTEN mutated). MW; molecular weight. (Lines; 741-742 in the revised manuscript).

3. Scale bars should be added to IF images

- As suggested, we have added scale bars to all IF images and these changes are detailed in associated figure legends.

4. The quality of the blots in Figure 3b are rather poor. Can they be improved? Are they representative of multiple experiments?

-Fig. 3b is a representative image of 3 independent experiments. Fig. 3b is now replaced with another representative immunoblot, which is displayed in better quality.

Reviewer #1 (Remarks to the Author)

This resubmitted manuscript deals with one major issue of GBM therapy, that being GBM's resistance to genotoxic drugs. In generating a XRCC4/P53 deleted mouse model driven by hGFAP-Cre, it was found that high-grade astrocytomas, specifically GBMs, develop among aged animals. Among these tumors, PTEN mRNA was reduced and the protein was aberrantly localized which was found to be dependent on its phosphorylation status. This was resultant of elevated CK2 expression in these tumors. Reactivating PTEN in vitro (relocalizing it to the nucleus) via inhibiting CK2, showed that genotoxic drug resistance can be obviated, thereby showing the therapeutic potential of a CK2 inhibitor that could target GBM cells that are resistant to DNA damage inducing drugs. Additionally, several PTEN mutants were generated to show that PTEN acts as a sensor to sensitize cells to DNA damage and that it is the nuclear localization of PTEN that promotes its DNA damage response activities. Lastly, it was shown that PTEN localization and subsequent DNA damage responsiveness is highly dependent on p53 status. This manuscript made some important findings regarding NHEJ, PTEN, and p53 in GBM with potential clinical application.

In reviewing this resubmission, all initial concerns were addressed and clarity was given to those parts that were previously either confusing, contradicting, or poorly worded. In light of the authors revisions, it would be recommended to accept without further revision.

Reviewer #2 (Remarks to the Author)

Most of my comments on the manuscript by Kang et al. have been addressed satisfactorily. However, some concerns remain:

2-1. Fig. 4. The competitive CK2 inhibitor, TBB, reduced CK2 and PTEN protein levels and reduced PTEN phosphorylation, resulting in reduced p-AKT and pPRAS40 levels. Despite the clear reduction in PTEN level, downstream AKT signaling is reduced, which would be consistent with enhanced PTEN activity. Yet, PTEN protein is predominantly localized in the nucleus in GXP GBM3 cells (Fig. 4b), which is not consistent with reduced p-AKT levels. This should be explained.

-We know that in the GBMs in our model, PTEN mRNA levels are low but the PTEN protein levels are comparable to neural stem cells, because of phosphorylation by CK2. TBB, by inhibiting CK2, results in the dephosphorylation and destabilisation of PTEN, which while transcriptionally reduced, is now catalytically active, able to localise to the nucleus in response to DNA damage and block PI3K/AKT signaling. To better clarify our findings, we have modified the text within lines; 224-228 to:

"Therefore, to test if pathological CK2 activity and PTEN subcellular distribution are coordinately regulated, we treated GXP GBMs, T98G and control NS with TBB, which by competitively inhibiting CK2 kinase activity, should reduce CK2-mediated PTEN phosphorylation and restore PTEN's protein turnover rate (Supplementary Fig. 2e,f) to reflect its physiologic reduced mRNA levels in GXP GBMs (Supplementary Fig. 2a)." (Underlined text here describes the key changes in the text in this section).

Explanation for the reduction in p-AKT levels is detailed within highlighted text within lines; 229-236, which we underline and detail below:

"Indeed, immunoblot analysis revealed that TBB reduced CK2, total PTEN and p-PTEN protein levels, and despite its reduced expression, restored PTEN's catalytic activity and ability to negatively regulate PI3K/AKT signaling, the latter shown by the reduction in p-AKT and p-PRAS40 in GXP GBMs (Fig. 4a).

A modest change in PTEN turnover was observed (Fig. S2e). These data are not convincing. The PTEN blots (-TBB) are overexposed. Given non-linearity of ECL, quantification of these blots is virtually impossible. It seems unlikely that the modest – if any – increase in turnover of PTEN causes the dramatic reduction in PTEN levels (Fig. 4a).

The reduction in PTEN protein levels and phosphorylation is clear (Fig. 4a). It is unlikely that reduced PTEN protein and phosphorylation levels directly cause reduced pAkt and pPRAS levels (the opposite is to be expected). Alternative pathways must be involved, particularly because PTEN predominantly localized to the nucleus, away from pAkt. This should be explained/addressed by the authors.

3-1. Fig. 5. CK2beta expression in GXP NS cells was modest (Fig. 5a), but had profound effects on cell proliferation (Fig. 5b-e).

-Data in Fig. 5 describe the effects of CK2 β overexpression, not endogenous CK2, and show that the overexpression of CK2 β profoundly increases the proliferation of GXP NS (XRCC4/p53-deficient neural stem cells), comparable to the levels observed for GXP GBMs.

The top panel of Fig. 5a shows high levels of endogenous CK2beta. Given this high level of endogenous CK2beta, the results in Fig. 5 are highly surprising. Transfection of CK2beta only leads to a modest increase in CK2beta levels, yet profound effects on PTEN phosphorylation and downstream effects. Is the transfected CK2beta distinct from endogenous CK2beta? Is transfected CK2beta constitutively active as opposed to endogenous CK2beta? This should be explained.

We thank Reviewer #1 for recommending acceptance of our revised manuscript without further revision. We further thank Reviewer #2 for their further careful consideration of our responses and additional comments, which have made every attempt to address in our point-by-point reply below. Within the revised manuscript, all changes have been highlighted using “Track Changes”.

Response to Reviewer #1:

Reviewer #1 (Remarks to the Author):

In reviewing this resubmission, all initial concerns were addressed and clarity was given to those parts that were previously either confusing, contradicting, or poorly worded. In light of the authors revisions, it would be recommended to accept without further revision.

-We thank reviewer #1 for his/her positive comments and all the support for our study.

Response to Reviewer #2:

Reviewer #2 (Remarks to the Author): We thank Reviewer #2 for their critical review of our overall data. We have made every effort below and as necessary, in the manuscript, to address additional concerns raised in this revision, which we hope will be considered satisfactory. Reviewer 2’s additional comments are bolded, and comments inclusive of our original responses, bolded and italicized, with key questions underlined. Our responses are in unbolded black or blue lettering following Reviewer #2’s complete comments.

2-1. Fig. 4. The competitive CK2 inhibitor, TBB, reduced CK2 and PTEN protein levels and reduced PTEN phosphorylation, resulting in reduced p-AKT and p-PRAS40 levels. Despite the clear reduction in PTEN level, downstream AKT signaling is reduced, which would be consistent with enhanced PTEN activity. Yet, PTEN protein is predominantly localized in the nucleus in GXP GBM3 cells (Fig. 4b), which is not consistent with reduced p-AKT levels. This should be explained.

-We know that in the GBMs in our model, PTEN mRNA levels are low but the PTEN protein levels are comparable to neural stem cells, because of phosphorylation by CK2. TBB, by inhibiting CK2, results in the dephosphorylation and destabilisation of PTEN, which while transcriptionally reduced, is now catalytically active, able to localise to the nucleus in response to DNA damage and block PI3K/AKT signaling. To better clarify our findings, we have modified the text within lines; 224-228

to: “Therefore, to test if pathological CK2 activity and PTEN subcellular distribution are coordinately regulated, we treated GXP GBMs, T98G and control NS with TBB, which by competitively inhibiting CK2 kinase activity, should reduce CK2-mediated PTEN phosphorylation and restore PTEN’s protein turnover rate (Supplementary Fig. 2e,f) to reflect its physiologic reduced mRNA levels in GXP GBMs (Supplementary Fig. 2a).” (Underlined text here describes the key changes in the text in this section).

Explanation for the reduction in p-AKT levels is detailed within highlighted text within lines; 229-236, which we underline and detail below: “Indeed, immunoblot analysis revealed that TBB reduced CK2, total PTEN and p-PTEN protein levels, and despite its reduced expression, restored PTEN’s catalytic activity and ability to negatively regulate PI3K/AKT signaling, the latter shown by the reduction in p- AKT and p-PRAS40 in GXP GBMs (Fig. 4a).

A modest change in PTEN turnover was observed (Fig. S2e). These data are not convincing. The PTEN blots (-TBB) are overexposed. Given non-linearity of ECL, quantification of these blots is virtually impossible. It seems unlikely that the modest – if any – increase in turnover of PTEN causes the dramatic reduction in PTEN levels (Fig. 4a).

The reduction in PTEN protein levels and phosphorylation is clear (Fig. 4a). It is unlikely that reduced PTEN protein and phosphorylation levels directly cause reduced p-AKT and p-PRAS levels (the opposite is to be expected). Alternative pathways must be involved, particularly because PTEN predominantly localized to the nucleus, away from p-Akt. This should be explained/addressed by the authors.

-Here, our response to Reviewer #2's original comments above was not deemed to be sufficient. We ask Reviewer 2 to consider and compare the PTEN subcellular fractionation data for GXP GBM3 (+TBB and TBB+DOX panel) in Supplementary Fig. 4b that they had requested in their original comments, to the PTEN subcellular fractionation data (+DOX panels for GXP GBM3) in Fig. 3b, and anti-PTEN immunofluorescence (IF) staining in Fig. 4b (bottom: + DOX and +TBB+DOX conditions). We believe that comparison of these 3 pieces of data addresses this question: Data from Fig. 3b provided evidence that PTEN in GXP GBM3 is resistant to DOX-induced DNA damage, as evidenced by its predominant cytoplasmic expression/localization in both -DOX and +DOX conditions. Supplementary Fig. 4b and Fig. 4b, which provide evidence that TBB treatment renders PTEN in GXP GBM3 responsive to DOX-induced DNA damage, show that while the nuclear fraction of PTEN is increased in TBB-treated GXP GBM3 cells, a substantial fraction of PTEN still resides in the cytoplasm with TBB treatment alone. This PTEN level is presumably and possibly sufficient to account for the reduced p-AKT and p-PRAS40 levels.

-As requested by Reviewer 2, immunoblots for PTEN turnover of GXP GBM2 and 3 are replaced with lower exposure (Supplementary Fig 2e,f). We believe these replaced blots address the inhibition of CK2-mediated PTEN phosphorylation restores the turnover rate of PTEN leading to the reduction of PTEN.

-Reviewer 2 further suggested a consideration of alternative mechanisms that might cause the reduction in p-AKT and p-PRAS40 levels. In this regard, while it is generally believed reduction in AKT activation with TBB treatment is a result of reconstitution of PTEN activity by CK2 inhibition using TBB reduces AKT activation, CK2 reportedly might also directly reduce AKT activation. We cite below examples of publications (lettering in blue) that describe the impact of CK2 inhibition using TBB treatment:

1. General roles:

In restoration of PTEN turnover rate. Citation: Silva, A., Yunes, J.A., Cardoso, B.A., Martins, L.R., Jotta, P.Y., Abecasis, M., Nowill, A.E., Leslie, N.R., Cardoso, A.A., and Barata, J.T. (2008). PTEN posttranslational inactivation and hyperactivation of the PI3K/Akt pathway sustain primary T cell leukemia viability. *J Clin Invest* 118, 3762-3774.

Reducing total PTEN protein levels. Citation: Patsoukis, N., Li, L., Sari, D., Petkova, V., and Boussiotis, V.A. (2013). PD-1 increases PTEN phosphatase activity while decreasing PTEN protein stability by inhibiting casein kinase 2. *Molecular and cellular biology* 33, 3091-3098.

Reducing AKT phosphorylation via reconstitution of PTEN activity. Citation: Shehata, M., Schnabl, S., Demirtas, D., Hilgarth, M., Hubmann, R., Ponath, E., Badrnya, S., Lehner, C., Hoelbl, A., Duechler, M., et al. (2010). Reconstitution of PTEN activity by CK2 inhibitors and interference with the PI3-K/Akt cascade counteract the antiapoptotic effect of human stromal cells in chronic lymphocytic leukemia. *Blood* 116, 2513-2521.

2. Alternative mechanisms:

CK2 inhibition reportedly may directly reduce AKT activation. Citation: Parker, R., Clifton-Bligh, R., and Molloy, M.P. (2014). Phosphoproteomics of MAPK inhibition in BRAF-mutated cells and a role for the lethal synergism of dual BRAF and CK2 inhibition. *Molecular cancer therapeutics* 13, 1894-1906.

Alternate PTEN-PI3K-independent mechanisms: Contributions from kinases not reliant on changes to PTEN, suggested as possible alternative mechanisms that modulate AKT.

- mTORC2. Citation: Bozulic, L., Surucu, B., Hynx, D., and Hemmings, B.A. (2008). PKB α /Akt1 acts downstream of DNA-PK in the DNA double-strand break response and promotes survival. *Molecular cell* 30, 203-213.

-DNA-PK. Citation: Feng, J., Park, J., Cron, P., Hess, D., and Hemmings, B.A. (2004). Identification of a PKB/Akt hydrophobic motif Ser-473 kinase as DNA-dependent protein kinase. *The Journal of biological chemistry* 279, 41189-41196.

-Integrin-linked kinase (ILK). Citation: Persad, S., Attwell, S., Gray, V., Delcommenne, M., Troussard, A., Sanghera, J., and Dedhar, S. (2000). Inhibition of integrin-linked kinase (ILK) suppresses activation of protein kinase B/Akt and induces cell cycle arrest and apoptosis of PTEN- mutant prostate cancer cells. *Proceedings of the National Academy of Sciences of the United States of America* 97, 3207-3212.

-Mitogen-activated protein kinase-activated protein kinase-2 (MAPKAPK2). Citation: Chan, C.H., Jo, U., Kohrman, A., Rezaeian, A.H., Chou, P.C., Logothetis, C., and Lin, H.K. (2014). Posttranslational regulation of Akt in human cancer. *Cell Biosci* 4, 59.

-To better clarify the data in regards to the findings in this section, we have modified Lines; 245- 258 in the revised manuscript as follows:

“Indeed, immunoblot analysis revealed that TBB reduced CK2, total PTEN and p-PTEN protein levels, and consequently decreased p-AKT and p-PRAS40 in GXP GBMs (Fig. 4a). This latter effect is most likely a direct consequence of negative regulation of PI3K/AKT signaling by TBB-restored catalytically active but reduced PTEN levels, although alternative mechanisms including a direct impact of TBB on AKT activation have been proposed^{41, 42}. By IF and nuclear/cytoplasmic fractionation, TBB increased PTEN nuclear expression, although a substantial fraction still resided in the cytoplasm, and further restored DNA damage induced nuclear expression of the remaining PTEN protein in GXP GBMs and T98G, while nuclear PTEN level was only partially restored in GXP NS, and minimally in WT NS (Fig. 4b, Supplementary Fig. 4a,b).”

3-1. Fig. 5. CK2beta expression in GXP NS cells was modest (Fig. 5a), but had profound effects on cell proliferation (Fig. 5b-e). Data in Fig. 5 describe the effects of CK2β overexpression, not endogenous CK2, and show that the overexpression of CK2β profoundly increases the proliferation of GXP NS (XRCC4/p53-deficient neural stem cells), comparable to the levels observed for GXP GBMs. The top panel of Fig. 5a shows high levels of endogenous CK2beta. Given this high level of endogenous CK2beta, the results in Fig. 5 are highly surprising. Transfection of CK2beta only leads to a modest increase in CK2beta levels, yet profound effects on PTEN phosphorylation and downstream effects. Is the transfected CK2beta distinct from endogenous CK2beta? Is transfected CK2beta constitutively active as opposed to endogenous CK2beta? This should be explained.

-Here, we ask Reviewer #2 to re-examine Fig. 5a, and look at the panel directly below the top panel, which shows the expression level of HA-tagged CK2β protein.

-The levels of endogenous CK2β protein expression in GXP NS compared to GXP GBMs, for which we show short and long exposures in Fig. 2d (shown in panels below PTEN and p-PTEN), does not induce p-PTEN to levels expressed in the GBMs. This suggests the endogenous CK2β in GXP NS is not sufficient to induce the profound downstream effects that occur as consequence of inhibiting PTEN.

-The transfected CK2β is constitutively expressed under the Cytomegalovirus (CMV) promoter and HA-tagged, as shown in the second panel of Fig. 5a. We believe this constitutive expression explains the profound effects on PTEN phosphorylation and downstream effects.